# Fusion of 2DGC-MS, HPLC-MS and Sensory Data to Assist Decision-Making in the Marketing of International Monovarietal Chardonnay and Sauvignon Blanc Wines

**DOI:** 10.3390/foods11213458

**Published:** 2022-10-31

**Authors:** Simone Poggesi, Aakriti Darnal, Adriana Teresa Ceci, Edoardo Longo, Leonardo Vanzo, Tanja Mimmo, Emanuele Boselli

**Affiliations:** 1Oenolab, NOI Techpark Alto Adige/Südtirol, Via A. Volta 13B, 39100 Bolzano, Italy; 2Faculty of Science and Technology, Free University of Bozen-Bolzano, Piazza Università 5, 39100 Bolzano, Italy

**Keywords:** MRATA (Modified Rate-All-That-Apply) sensory analysis, HS-SPME-GC × GC-TOF/MS, HPLC-MS, chardonnay, Sauvignon Blanc, cyclic proanthocyanidins, international wines

## Abstract

Monovarietal wines produced in different wine-growing areas may have completely different sensory profiles. As a result, they may be suitable for sale in different regions, depending on local preferences. Better insight into the sensory and chemical profiles of these wines can be helpful in further optimizing commercial strategies and matching supply and demand, which is the main challenge for global wine traders. The training of dedicated sensory panels, together with the correlation of the evaluated attributes with chemical parameters, followed by validation of the obtained models, may yield an improved picture of the overall features associated with products from a specific region. Eighteen samples of international Chardonnay and eighteen samples of international Sauvignon Blanc wines were collected from nine world origins (Northern Italy, Southern Italy, Chile, Argentina, New Zealand, Australia, and South Africa). The overall quality judgement (OQJ) and the sensory attributes were evaluated by a panel trained with a MRATA (Modified Rate-All-That-Apply) method. Moreover, volatile compounds were analysed by HS-SPME-GC × GC-ToF/MS and the phenolic composition, including proanthocyanidins, was determined using HPLC-QqQ/MS. The processing of the data using different multivariate analysis methods, such as multiple factor analysis (MFA), was essential to gain insight into the quality of the samples. The profile of cyclic and non-cyclic oligomeric proanthocyanidins was found to be substantially dependent on the grape variety used in the wines (varietal markers), despite the country of origin of the wine influencing it to a limited extent. The results from the same samples analysed by a sensory panel from Germany and ours were qualitatively compared, highlighting the presence of potential factors inherent to the panels themselves that could influence the different judgments and quality classification of the wines. Consequently, the combination of sensory and chemical analysis, by means of the application of multivariate statistical methods presented in this study proves to be a powerful tool for a deeper and more comprehensive understanding of the quality of the wines under investigation. Overall quality was described as a combination of the sensory attributes, according to the perception process. The attributes were in turn described based on the chemical profiles, which were determined independently by analytical techniques. Eventually, this approach can be very useful not only for basic research on wine quality but also as a tool to aid business-related decision-making activities of wineries and wine traders and to create models that can aid the refinement of marketing strategies.

## 1. Introduction

Wine quality is an issue that has been debated for a long time. As for other food products, there may be different angles from which “quality” can be defined and assessed. Charters and Pettigrew defined quality as a multidimensional concept that includes a wide range of intrinsic and extrinsic factors [1]. Moskowitz (1995) defined quality as a situation-specific concept that is fluid and dynamic over time [2]. In the sensory analysis applied in the wine industry, sensory stimuli are usually divided into colour sensation, aroma, mouthfeel (trigeminal nerve), and flavour. This classification, based on physical perception patterns, has also been used to train sensory panels and provide a track for consumer tests in recent decades [3]. However, when consumers or panellists evaluate wine, there are multiple interactions between sensory properties, nerve system physiological structures, and consciousness, and it is difficult to fully identify the role of all these variables [4,5]. As a result, the interactions between different factors define the quality of the product perceived by the consumers and accordingly influence their preferences. Multiple interacting factors make a complete standardization of the approaches to define sensory quality quite difficult. Recent attempts have included the integration of sensory and chemical profiles, with the aim of explaining trends in sensory evaluation based on wine composition, applying chemometric approaches [6]. Of course, a limitation in the samples available for such a study could pose an issue in terms of the representativeness of the overall significance of the results. Factors that might influence the general discussion about the quality of wines from the same grape variety could be applied in winemaking practices, such as terroir, vintage, etc.

From the point of view of wine quality classification, this work aims to profile the main sensory characteristics of monovarietal Sauvignon Blanc and Chardonnay wines from nine world origins (Northern Italy, Southern Italy, Chile, Argentina, New Zealand, Australia, and South Africa), quantitatively to describe the sensory attributes most influencing the assigned overall quality, and to determine the relationships between the sensory descriptors and chemical profiles, by applying multivariate statistics. This research objective is not only of basic scientific importance but could be a useful tool to aid the business choices of a winery or wine trader, in addition to offering a parallel tool to support wine evaluations by panels of experts, which are commonly used in the wine market, as well as consumer tests.

Chardonnay is a grape variety that is documented to have originated in France, specifically in the Burgundy region around 500 years ago [7]. It is cultivated in 41 countries around the world with a reported addition of 210,000 hectares in 2017, and an estimated world average production of 882,000 hL of wine in 2017 [8]. The main producers are France, Italy, Spain, the United States, Australia, and Chile. For these reasons, Chardonnay can be considered one of the most important grape varieties for wine production. Chardonnay is characterised by buttery, tropical fruit, and citrus attributes due to the presence of odorants such as diacetyl, thiols, and esters above their aromatic thresholds, allowing for their perception [7]. Chardonnay, along with Pinot Blanc and Pinot Noir, is one of the most important varieties used to produce sparkling wines made by refermentation in the bottle (*champenoise* method). Notable examples can be found in France (Champagne area) and Italy (Trentino Alto-Adige and Franciacorta) [9].

Like Chardonnay, Sauvignon Blanc is an indigenous grape variety of the Loire Valley region (central France) [9]. It is considered one of the most aromatic varieties; its main descriptors are grapefruit, guava, cat urine, and passion fruit due to the content of thiol compounds, while methoxypyrazines give hints of herbaceous plants, green pepper, and tomato stems [10]. Nowadays, Sauvignon Blanc is considered one of the main international white grape varieties and it is grown in more than 15 countries, with an estimated world average production of 861,000 hl of wine in 2017 [8]. The most important production areas for this variety are France, New Zealand, California, Chile, South Africa, and Italy.

In this study, standard oenological parameters were determined in combination with volatile compounds analysed by two-dimensional gas chromatography (GC × GC-MS), phenolic compounds and proanthocyanidins by HPLC-MS, and sensory attribute profiles with a modified-rate-all-that-apply (MRATA) sensory test in monovarietal Sauvignon Blanc and Chardonnay wines coming from nine different origins. Based on the different analytical protocols, the quality was defined using different multivariate explorative and regression approaches. Analysis of variance (ANOVA), agglomerative hierarchical clustering (AHC), and multiple factor analysis (MFA) were used as explorative methods, while partial least squares regression (PLS-R), principal component regression (PCR), or multiple linear regression (MLR) were applied as regression approaches to understand which variables had a major impact on the sensory perceptions. In addition, the LC-MS profile of proanthocyanidins was studied to be applied as a varietal chemical marker, i.e., to investigate common profile fingerprints due to the variety and influences of non-varietal factors such as origin [11].

## 2. Material and Methods

### 2.1. Chardonnay and Sauvignon Blanc Wines

The wine samples were provided by FruitService GmbH (Bolzano, Italy) in 0.75 L bottles, in triplicate. All the samples were labelled as monovarietal wines from Chardonnay and Sauvignon grapes produced in 2020 in different countries. The unique code generated for each sample and commercial name of the wine are reported in Table 1 for Chardonnay and Table 2 for Sauvignon Blanc. The oenological parameters reported in the two following tables were analysed by the winery.

### 2.2. Climate of the Geographical Area

In Table 3, climate details for the different origins of the samples are given. There are only two samples from the northern hemisphere (Italian wines), while the other samples are from the southern hemisphere. As shown in Table 3, all areas have different climatic conditions and can be divided into three main climate types: oceanic climates, continental climates, and Mediterranean climates.

### 2.3. Sensory Analysis

For the sensory analysis of Chardonnay and Sauvignon Blanc wines, three phases (defining the sensory attributes, rapid training, and RATA test) each consisting of one round table, two training sessions (1 h each), and two MRATA sessions (modified rate all that apply [24]) were performed. Cysensy–an SQL binding sensory analysis web software developed in collaboration with the Computer Science faculty of the Free University of Bozen/Bolzano was used during the training and MRATA sessions.

The panellists were recruited voluntarily and included students and staff of the Free University of Bozen/Bolzano (Italy). Training and sensory testing were conducted in the same room and at the same time for one month. To limit the impact of location and time on the reproducibility of the sensory analysis, these parameters were maintained for all sessions.

The panellists for both Chardonnay (9 panellists, 90% males and 10% females, aged 26 ± 2 years old) and Sauvignon Blanc (10 panellists, 40% females and 60% males, aged 26 ± 5 years old) had a background in oenology. For both Chardonnay and Sauvignon Blanc, the total number of samples evaluated were 27: nine bottles were used to create the common vocabulary, while 18 bottles were used in the RATA quantitative assessment (9 origins × 2 replicates).

In the round table session, the panellists were asked to test each wine individually and share descriptors of visual, olfactory, and gustatory (taste and flavour) attributes on an interactive web whiteboard (Jamboard, Google, Netherlands). After each sample, the panel leader and panellists discussed and selected the most appropriate shared attribute for each wine type.

Then, the panellists were trained for those specific attributes (shown in Table 4). For both varieties, the first training session was specific to the odour attributes. Panellists were trained with reference standards [25] or natural standards if references were not available. The odour attributes of Chardonnay were “citrus” (lemon, grapefruit), “floral” (rosewater, green tea), “green vegetables” (green pepper), “stone fruit” (peach), and “tree fruit” (green apple).

On the contrary, for Sauvignon Blanc, the odour attributes were “tropical fruit” (passion fruit, mango), “stone fruit” (apricot, peach), “citrus” (grapefruit, lemon), “dried fruit” (nutty, almond), “woody”, “vanilla”, “green grass”, “tomato stem”, “green bell pepper”, “flowers” (wildflower, jasmine), and “honey”. The second training session was assigned to those odour attributes that the panellists did not recognise at a percentage higher than 75% during the descriptor test of the first training session, along with the taste and flavour attributes. For Chardonnay, “warmness”, “sourness” (citric, lactic, malic, and tartaric acid), and “bitterness” (caffeine) were selected for the taste attribute during the round table, and “citrus”, “pome tree fruit”, “herbaceous”, and “floral” for the flavour attributes. For Sauvignon Blanc, “warmness”, “sourness” (citric and tartaric acid), and “sweetness” were selected for the taste attributes and “citrus” and “woody” were selected for the flavour attributes.

Then, the panellists performed a MRATA in which they were asked to rate the intensity of each attribute from 0 to 5, where 0 was “does not apply”, 1 was “weak”, and 5 was “strong”.

One of the attributes was Overall Quality Judgment (OQJ), a descriptor that combines all three aspects of the sensory analysis (visual, olfactory, and gustatory), and should be considered as an objective evaluation of the overall quality of the product. The wine samples were analysed in duplicate in two different sessions. The bottles were opened 30 min before the session. An aliquot (30 mL) of wine was poured into ISO glasses labelled with a random 3-digit number. The tasting order of the wine samples was randomised using a complete Latin square randomization among the panellists to reduce carry-over effects [26]. In addition, a sensory test was also performed in an accredited laboratory for wine testing (DIN EN ISO/IEC 17025:2018) by a local German panel.

### 2.4. Method Optimization for HS-SPME

To increase the instrumental response towards the compounds that mainly define the typicity of Chardonnay and Sauvignon Blanc wine odours, solid-phase micro-extraction (SPME) temperature, incubation time, extraction time, and fibre composition were optimised by applying a central composite experimental design [27,28]. The optimisation was designed on three quantitative factors (extraction temperature, incubation time, and extraction time) and one qualitative factor (fiber type). Minitab (version 19. 2020.1, Minitab LLC, State College, PA, USA) was used to create the experimental design and calculate the models. In Table 5, the factor list used in the experimental design was reported. The optimiser tool of Minitab was used to combine the different models obtained for the different classes with the aim of improving the detection of the most-difficult-to-analyse compounds (i.e., especially the relatively polar volatile compounds, such as thiols and methoxypyrazines).

To study the effect of different factors, a model wine was prepared by adding several standard volatile compounds stocked in ethanol to a commercial wine (white Tavernello, Caviro winery, Faenza) from the same batch (main technological parameters, determined by MIURA analysis: acetic acid = below quantitation threshold, glucose + fructose = 8.1 g·L^−1^, L-tartaric acid = 0.7 g·L^−1^, L-lactic acid = 0.4 g·L^−1^, L-malic acid = 1.1 g·L^−1^, total polyphenols index (TPI–Folin) = 180 mg·L^−1^, total-SO_2_ = 94 mg·L^−1^, free-SO_2_ = 11 mg·L^−1^).

### 2.5. HS-SPME-GC × GC-ToF/MS Analysis of the Volatile Profiles

For sample preparation, 0.5 g of NaCl was placed into a 10-mL SPME glass vial. Then, a 4 mL wine sample was added to the vial. Next, 10 µL of 2-methyl-3-pentanol was added as an internal standard (from a stock of 1/50 of I.S. in ethanol). Then, the vial was closed with a perforable silicon screw-cap. For SPME extraction, a Stableflex SPME fiber 50/30 µm DVB/CAR/PDMS, 23 Ga, for the autosampler (SUPELCO, 595 North Harrison Road, Bellefonte, PA, 16823-0048, USA) was used. The solid-phase microextraction incubation time was 5 min and extraction time was 60 min. Both the extraction and incubation were under continuous heating (50 °C) and stirring (300 rpm). The SPME was performed with an LPAL3 GC autosampler equipped with a Peltier Stack, where the samples were kept at 4 °C before the analysis.

The fiber was preconditioned at 240 °C for 6 min before the analysis. The gas chromatographic method was adapted from previously published reports [6,29]. The separation was performed by comprehensive GC × GC on an instrument coupled with a Pegasus BT 4D time-of-flight mass spectrometer and equipped with a Flux^TM^ flow modulator (Leco Italy, Milano). The separation was performed at 1 mL/min (He carrier gas) in splitless mode, with 2 mL/min septum purge flow and 6 mL/min inlet purge flow. The inlet temperature was 240 °C. Each sample was analysed in GC × GC on a polar cross-bond PEG-phase MEGA-Wax Spirit 0.30 µm × 0.18 mm × 40 m (Mega S.r.l., Legnano, Italy) as the first dimension and a Rxi-17 Sil 0.10 µm × 0.10 mm × 1.2 mm (Restek S.r.l., Cernusco sul Naviglio, Italy) as the second dimension. The two columns were connected by a Flux^TM^ modulator (LECO), with a 15.92 psi auxiliary diverting helium flow. The second column was stored in a secondary oven inside (but thermally isolated from) the larger primary oven. The primary oven temperature program was 40 °C for 6 min, 40 to 180 °C at 3 °C/min, 180 to 240 °C at 10 °C/min, and 240 °C for 1 min. The secondary oven had a temperature 5 °C higher than the first oven, and accordingly, its temperature program was shifted to that of the same temperature difference. The GC × GC instrument was connected with the mass spectrometer by a transfer line kept at 250 °C. The mass spectrometer was operated with the following parameters: 0 s acquisition delay, 70 eV filament electron energy, 250 °C ion source temperature, 150 spectra/s acquisition rate, 32 kHz extraction frequency, and acquisition mass range = *m*/*z* 35–530. The mass spectrometer was tuned before starting a new sequence of analyses. After each analysis, each GC × GC chromatogram was reconstructed in 2D colour maps by ChromaTOF^®^ software by deconvolution. Automatic compound assignment was done by NIST 2017 (NIST MS Search 2.3) database comparison. Linear retention indexes were calculated from the first-dimension retention times against the injected series of linear fatty ethyl esters C4-C24 (even carbon saturated, Merck Life S.r.l., Milan, Italy).

The processing software ChromaTOF^®^ (LECO Corporation, Berlin, Germany) ver.2021 was used to process the chromatograms obtained from the bidimensional gas-chromatography. Target analyte finding (TAF) for specific compounds in both Chardonnay and Sauvignon Blanc was also used. The parameters used in the TAF methods are reported in Appendix A.

### 2.6. UHPLC-MS Analysis of the Phenolic Profiles

The LC-MS analysis of polyphenols was conducted on a UHPLC-QqQ/MS instrument (Agilent LC/TQ 6465 system) equipped with a 1260 Infinity II UHPLC with a quaternary pumps system, a 1260 Infinity II WR PDA detector, in series to an AJS ESI QqQ mass analyser. The chromatographic separations were conducted on a Poroshell 120, SB-C18 2.1 mm × 100 mm × 2.7 µm (Agilent Technologies Italia, Milan, Italy) kept at 30 °C at a 0.35 mL/min flow rate. The mobile phase consisted of A) 0.1% formic acid in degassed ultrapure water and B) 0.1% formic acid in acetonitrile. All used solvents and mobile phase additives were at MS grade. The operational conditions to analyse the Chardonnay wine samples were as follows: The gradient separation program was 1%B from 0 to 1.5 min, 1 to 30% B from 1.5 to 19 min, 30 to 99% B from 19 to 20 min, 99% B from 20 to 24 min, 99 to 1% B from 24 to 25 min, and 1% B from 25 to 30 min. The injection volume was 5 µL. The PDA detector was set to record the absorbance in the 200–700 nm wavelength range using a 4 s response time (1.25 Hz) and 4 nm slit width, with 1 nm spectrum steps. The MS detection was performed in ESI- ionization mode, with the following parameters: mass range = *m*/*z* 200–750, scan time = 500 ms, step size = 0.1 amu, fragmentor potential = 135 V, cell acceleration = 5 V, N_2_ gas temperature = 340 °C, N_2_ gas flow = 13 L/min, nebuliser pressure = 50 psi, sheath gas heater = 350 °C, sheath gas flow = 12 L/min, capillary voltage = −3500 V, nozzle voltage = −500 V.

The operational conditions to analyse the Sauvignon Blanc wine samples were as follows: The gradient separation program was 1%B from 0 to 2.5 min, 1 to 25% B from 2.5 to 50 min, 25 to 99% B from 50 to 51 min, 99% B from 51 to 55 min, 99 to 1% B from 55 to 56 min, and 1% B from 56 to 62 min. The injection volume was 5 µL. The PDA detector was set to record the absorbance in the 200–700 nm wavelength range using a 4 s response time (1.25 Hz) and 4 nm slit width, with 1 nm spectrum steps. The MS detection was performed in ESI- ionization mode, with the following parameters: mass range = *m*/*z* 100–1000, scan time = 500, step size = 0.1 amu, fragmentor potential = 130 V, cell acceleration = 5 V, N_2_ gas temperature = 260 °C, N_2_ gas flow = 4 L/min, nebuliser pressure = 35 psi, sheath gas heater = 300 °C, sheath gas flow = 12 L/min, capillary voltage = −2500 V, nozzle voltage = −2000 V. MS raw data were converted into mzData format with MassHunter (Agilent) and exported. The MzMine3 (http://mzmine.github.io/) application was employed for automatic alignment and pre-processing before statistical analysis. PDA retention times were corrected to match the MS retention times. Appendix A reports the workflow applied to obtain the dataset of phenols.

Tentative compound identification was conducted by full-scan mass spectrometry determination and the relative PDA λ_max_ assignment, to classify the compounds at least in a phenolic class, where a complete identification could not be achieved. Solutions of standard compounds (gallic acid, trans-caffeic acid, p-coumaric acid, trans-caftaric acid, (+)-catechin, (-)-epicatechin, protocatechuic acid, astilbin, kaempferol-3O-glucoside, and procyanidin B2) were analysed by standard injection, and their PDA spectra, MS/MS spectra, and retention times (min) were used as references. The tentative identification of phenolic compounds was reported in Appendix A. Targeted MS/MS fragmentation experiments (product ions monitoring-PRM) on selected ions were conducted with the following setup: source parameters as in MS1 analysis, acquisition mass range from *m*/*z* 25 to + 10 *m*/*z* from the selected precursor ion, scan time = 125 ms, fragmentor potential = 135 V, collision energy = 25 V, and cell accelerator 5 V. The results of the fragmentation experiments were reported in Appendix A.

Proanthocyanidins (PAC) were separately analysed on the same UHPLC-QqQ/MS instrument as for non-volatile phenols. The chromatographic separation was conducted in a Vertex Plus Eurosphere II (KNAUER, Berlin, Germany) column 4.6 mm × 250 mm × 5 µm with a precolumn, and kept at 30 °C. The separation was conducted at a 0.7 mL/min flow rate. The mobile phases consisted of A) 0.1% formic acid in degassed ultrapure water and B) 0.1% formic acid in acetonitrile. All solvents and mobile phase additives were at MS grade. The gradient separation program was as follows: 1% B from 0 to 2.5 min, 1 to 25% B from 2.5 to 50 min, 25 to 99% B from 50 to 51 min, 99% B from 51 to 55 min, 99 to 1% B from 55 to 56 min, and 1% B from 56 to 62 min. The injection volume was 5 µL. The mass spectrometer operated in ESI^+^ mode. The detection of the proanthocyanidins was performed in single-ion-monitoring (SIM) mode. In Appendix A, the SIM parameters for both Chardonnay and Sauvignon Blanc are listed. The parameters for mass spectrometric detection were the following: fragmentor potential = 135 V, cell acceleration = 5 V, N_2_ gas temperature = 230 °C, N_2_ gas flow = 8 L/min, nebuliser pressure = 20 psi, sheath gas heater = 300 °C, sheath gas flow = 10 L/min, capillary voltage = 3000 V, nozzle voltage = 2000 V. Integrated peaks were directly downloaded from Mass Hunter and aligned manually.

### 2.7. Statistical Analysis

The statistical analysis was performed using the XLSTAT add-on for Excel (Addinsoft, Paris, France). The datasets for the two different varieties were basic oenological parameters, sensory analysis (visual, olfactory, gustatory, overall quality judgment), volatile compounds, non-volatile phenolic compounds, and proanthocyanidins.

One-way ANOVA (analysis of variance) and the Tukey HSD (honestly significant difference) *post-hoc* test were used to identify the most significant variables differentiating the samples. The sensory analysis dataset was also investigated by Agglomerative Hierarchical Clustering (AHC) method. Sensory datasets were computed using the dissimilarity and Euclidian distance coefficient (XLSTAT help, 2022, https://www.xlstat.com/en, accessed on 31 May 2022).

To explore the relationship between experimental variables and factors, Multiple Factor Analysis (MFA) was applied. This statistical method allows the analysis of a dataset containing different groups of variables (both quantitative and qualitative) [30]. The MFA methodology is divided into two phases: performing Principal Component Analysis [31] (for quantitative data) and Multiple Correspondence Analysis (for qualitative tables) or Correspondence Analysis (for frequency tables). Then, the MFA weighs the sets of variables normalizing the highest axial inertia of each set. The axial inertia is a measure of the weighted spread of the point in a multidimensional space (XLSTAT help, 2022). The MFA has been performed with 7 sets of continuous variables: wine basic parameters, sensory analysis divided by the different modes of perception (colour, aroma, gustatory-flavour, overall quality judgment), volatile compounds, and phenolic compounds.

Partial Least Square Regression (PLS-R) has been used to correlate the descriptor variables with the overall quality judgment to understand which descriptors most affected the quality of the product [6]. The Q^2^ cumulative index represents the goodness of fit in the prediction quality of the model and was calculated using the following (Formula (1))
(1)$$Q^{2}\mathrm{cum}\;\left(h\right)=1-\overset{h}{\underset{j=1}{\prod }}\frac{{\mathrm{PRESS}}_{kj}}{{\mathrm{SSE}}_{k\left(j-1\right)}}$$
where “h” is the number of components used in the regression model, “*j*” is the observation and involves the predicted residual error sum of square (PRESS) statistics and the Sum of Squares Errors (SSE). Other important indexes are R^2^Y, R^2^X (which correspond to the correlation with the components between the dependent (*Y*) and the explanatory (*X*) variables), and the VIP (variables importance in projection). The VIP index explains the importance that each variable has in the model on the latent factors (Formula (2))
(2)$${\mathrm{VIP}}_{\mathrm{hj}}=\sqrt{\frac{p}{\sum_{i=1}^{h}Rd\left(Y,t_{i}\right)}\cdot \overset{h}{\underset{i=1}{\sum }}w_{ij}^{2}Rd\left(Y,t_{i}\right)}$$

The PLS2 regression method was used to identify which volatile compounds affected the olfactory and flavour sensory attributes. Also, PLS2 regression was used to understand which non-volatile phenolic compounds affected the visual and gustatory sensory attributes.

The Multiple Linear Regression (MLR) method was used to calculate the regression for the gustatory variables on the dataset of Chardonnay wines. Multiple linear regression is a statistical feature that models the values of a quantitative dependent variable *Y* through a linear combination of quantitative explanatory variables *X*. The determinist model is reported below (Formula (3))
(3)$$y_{i}=\beta_{0}+\overset{p}{\underset{i=1}{\sum }}\beta_{j}x_{ij}+\epsilon_{i}$$
where “*y_i_*” is the value observed for the dependent variables (observation *i*),“*x_ij_*” is the value taken by variable *j* for observation *I*, and “*ϵ_i_*” is the error of the model.

## 3. Results

### 3.1. Analysis of Variance

#### 3.1.1. Chardonnay Sensory Data

Figure 1 shows the significant variables determined by one-way ANOVA for the Chardonnay sensory data. The significant visual descriptors found were “Green colour” and “Yellow colour”, as shown in Figure 1A. “Citrus fruit” aroma, “Warmness” taste, “Tropical fruit” and “Woody” flavours are shown in Figure 1B. The computed LS means and the groups (indicated by the alphabetical letters) obtained by one-way ANOVA and Tukey HSD test (α = 0.05) are reported in Appendix A. The significant variables for the volatile compounds and phenolic compounds are reported in Appendix A, respectively. Appendix A report the PCA observations and bi-plots, for volatile compounds and non-volatile phenolic compounds of Chardonnay wines, respectively. Figure 1A for the significant colour attributes (“Yellow colour” and Green colour”) showed that ARG_ME had the highest intensity in the “Yellow colour” and lowest in the “Green colour”. This wine was followed by RSA_WC and CLE_CV for the “Yellow colour” intensity, although Tukey’s test did not show a significant difference between the samples for the “Yellow colour”. Instead, for the “Green colour”, as mentioned above, the CLE-CV wine showed the lowest intensity. Moreover, Tukey’s test grouped these samples with the letter “b”, while the other samples were grouped as “ab”, thus there were no significant differences between these samples, but just a trend. Regarding the “Citrus fruit-a” aroma attribute (Figure 1B), RSA_CO and RSA_WC showed the highest level for this descriptor, whereas ARG_ME and ITA_SP were characterised by the lowest one. These samples were divided into two groups by Tukey’s test: the first two samples were divided into one group and the other two into another group, which confirmed the significant difference between the samples. For “Warmness”, the highest samples were NZL_MA, RSA_WC, and CLE_CV, which were grouped by Tukey’s test. The lowest sample was ITA_SP, which was included in another group showing a significant difference. The “Tropical fruit-f” descriptor did not show significant differences in Tukey’s test; only one group is shown. However, the samples with the highest level were AUS_SE and CLE_CV. For “Woody-f”, the results differentiated many significant groups using the Tukey test; the samples with the highest level were NZL_MA and ARG_ME, whereas the samples with the lowest level were ITA_SP and ITA_VF, which were divided into two different groups. This result showed how the Italian wines were poor in the “Woody-f” descriptor: the use of wood for the Chardonnay wines was not applied to these Italian Chardonnay wines.

#### 3.1.2. Sauvignon Blanc Sensory Data

##### Analysis of Variance

One-way ANOVA reported in Figure 2 shows the significant variables in the sensory data of Sauvignon Blanc wines. The significant variables were “Green colour” and “Yellow colour” (Figure 2A), “Tomato stem” and “Green bell pepper” aromas (Figure 2B), and the “overall quality judgment (OQJ)” (Figure 2C). In Appendix A, the LS mean and Tukey’s HSD post-hoc test results are shown. One-way ANOVA and Tukey’s HSD test for the volatile compounds and non-volatile phenolic compounds, respectively, are shown in Appendix A. Also, PCA observations and the bi-plot for volatile compounds and non-volatile compounds, respectively, on Sauvignon Blanc wines were reported in Appendix A.

The colours “Green colour” and “Yellow colour” (shown in Figure 2A) showed a similar trend. “Green colour” did not show significant differences in Tukey’s test. However, the samples with the highest value of this descriptor were RSA_WC, while the lowest ones were ITA_SP and NZL_MR. The “Yellow colour” showed 4 different groups by Tukey’s test, where ARG_ME was the sample with the highest level (group “a”), and the lowest sample was NZL_MR (group “c”). Regarding the significant aroma descriptors (shown in Figure 2B), “Tomato stem” and “Green bell pepper” were similar in the samples. Only NZL_MA showed differences as “Tomato stem” was higher than “Green bell pepper”. The highest samples were RSA_CO and the lowest was ITA_SP for both descriptors. “OQJ” (shown in Figure 2C) was divided by Tukey’s test into three groups. The lowest sample was ARG_ME, which was significantly different from all the other samples.

##### Agglomerative Hierarchical Cluster Analysis (AHC)

AHC results are only briefly discussed as qualitative analysis will be comprehensively discussed with the MFA. However, AHC was able to cluster almost all the origins. Appendix A showed two principal clusters: The first cluster grouped ARG and RSA origins, while the second grouped the other origins in two main sub-groups, NZL_MR and AUS_SE + ITA_VF. Appendix A shows three main clusters of Chardonnay wines. The first one with the ITA SP + ITA VF origin, the second one with ARG_ME, different from the other wines, and the last one clustered the other origins. Notably, the inclusion of overall quality in the parameters’ dataset greatly improved the separation of the samples by country of origin.

### 3.2. Multiple Factor Analysis

#### 3.2.1. Chardonnay Wines

Figure 3 shows the MFA for the Chardonnay wines computed with the dataset of basic oenological parameters, sensory analysis, volatile compounds, and phenolic compounds. MFA was used to fuse the different datasets into a single statistical computation, to give a generalised overview of the correlations between the variables and of the trends among the observations. The partial axes built on the first two components shown in Figure 3A accounted for 38% of the total variance. From Figure 3A,B, clear separation among wines from Italy can be seen due to the contribution of the phenolic compounds-F2, basic oenological parameters-F1, taste and flavour-F1, and visual-F2. Similarly, volatile compounds-F2 mostly contributed to the separation of wines from Argentina. Phenolic compounds-F1 and olfactory attributes-F2 correlated with the wines from South Africa, and basic oenological parameters-F2, olfactory-F1, and visual-F2 contributed to the separation of wines from Australia. The wines from New Zealand were also well separated due to the contribution of taste and flavour-F2, volatile compounds-F1, and OQJ. Figure 3C shows that pH and the residual sugar were anti-correlated. The distribution of the observations shows that wines from New Zealand (Marlborough) and South Africa correlated with the pH, which was higher in both wines, and anti-correlated with the residual sugars; while wines from Australia and New Zealand (Hawkes Bay) correlated with the titratable acidity, which was highest among the wines. The loadings plot in Figure 3B–E shows that the OQJ correlated with “flowers”, “citrus fruit” (aromas), and “tropical fruit” (flavour). These are also the attributes that characterised wines from Australia. Additionally, volatile compounds XVI, acetic acid-2-phenylethyl acetate (XXVI), naphthalene, decahydro-4a-methyl-1-methylene-7-(1-methylethylidene) (XXXIII), and butanedioic acid, diethyl ester (XXIII) showed a correlation with OQJ. Notably, the ester 2-phenylethyl acetate is responsible for fresh and floral aromas in wine [32]; butanedioic acid, diethyl ester is shown to give a fruity odour perception [33]. Finally, naphthalene, decahydro-4a-methyl-1-methylene-7-(1-methylethylidene) is shown in association with woody perceptions [34].

The Italian wines were separated from other origins due to the volatile compounds (XXVIII), (XXVII), (IX), (XXII), and (V). Wines from New Zealand (Hawkes Bay) were characterised by “green colour”, and wines from Chile by the olfactory “stone fruit”. From Figure 3C, the correlations of the distributions of specific phenolic compounds with specific sample groups are highlighted; wines from Argentina showed higher contributions of x.11 (*m*/*z* 333.1 at 1.3 min) and x.96 (*m*/*z* 311.1 at 6.9 min); wines from South Africa showed higher contributions of the ion x.169 (*m*/*z* 429.1 at 11.5 min); wines from New Zealand showed higher contributions of x.68 (*m*/*z* 153.0 at 5.5 min) and x.115 (*m*/*z* 284 at 8.1 min). The species that particularly characterised the wines from Italy were x.5 (*m*/*z* 205.1 at 1.0 min), x.75 (*m*/*z* 368.1 at 5.8 min), x.69 (*m*/*z* 315.1 at 5.6 min) and x.13 (*m*/*z* 405.1 at 1.5 min). A tentative identification of the most relevant phenolic compounds was conducted by comparing our full MS and fragmentations (MS^2^) results with the current pieces of literature.

The ion x.11 was characterised by *m*/*z* 333.1. Although the precursor ion was *m*/*z* 347.1, it was possible to observe its in-source fragmentation at *m*/*z* 333.1. It was hypothesised that it could be a dimer of quinic acid (with its visible in-source fragment *m*/*z* 191), generated by ester condensation reactions. A reason for the in-source fragmentation for this species could be an applied too high ionization potential; the mass spectrum of *m*/*z* 331 showed the fragment *m*/*z* 191, backing up our hypothesis. Moreover, the fragmentation spectrum of *m*/*z* 333.1 also showed *m*/*z* 217, *m*/*z* 173 and *m*/*z* 111 as fragments [35]. Then, it was speculated that ion x.11 at *m*/*z* 333.1 and MS^2^ product ions at *m*/*z* 173, *m*/*z* 111, *m*/*z* 155, and *m*/*z* 67 could be a derivative of citric acid [36]. Ion x.96 (*m*/*z* 311.1 at 6.9 min) was characterised in MS^2^ by *m*/*z* 149 (tartaric acid) and *m*/*z* 135 (decarboxylated caffeic acid) and accordingly was assigned to caffeoyltartaric acid (caftaric acid) [37]. Ion x.169 showed as precursor ion *m*/*z* 429.1 that was hardly identified. However, the mass spectra contained fragments at *m*/*z* 366 and *m*/*z* 204, which were possibly obtained by a neutral loss of glucose residues [38]. It was hypothesised that the fragment at *m*/*z* 204 was a derivative of the dimer of N-acetyl-glucosamine bound to a glucose moiety. Moreover, considering the found *m*/*z* 62 fragment, and the related neutral losses (Da), it was hypothesised that this was due to an adduct with ammonium formate (although no ammonium salt was added to the mobile phases), or a complex with copper. Ion x.68 showed a negative parent ion at *m*/*z* 153.0 (at 5.5 min) and a fragmentation pattern characterised by *m*/*z* 109 and *m*/*z* 108, which could be characteristic of protocatechuic acid [39]. The full MS spectrum of the ion x.115 showed as fragment ions *m*/*z* 163 (coumaric acid) and *m*/*z* 149 (tartaric acid). As reported by Šuković et al. [40], x.115 was tentatively identified as coumaryltartaric (coutaric acid); its precursor ion at *m*/*z* 295 was also identified as in co-elution, supporting this hypothesis. The product ion at *m*/*z* 119 results therefore from the loss of a CO_2_ unit (44 Da) from *m*/*z* 163 [40].

Furthermore, the ion *m*/*z* 284 has been reported as the deprotonated radical ion of kaempferol; nevertheless, the most common fragments of kaempferol were not found in the MS^2^ spectrum [41]. The ion *m*/*z* 119 has been found as a fragment in the databases for kaempferol (massbank.eu) [42]. x.69 showed a negative parent ion at *m*/*z* 315.1 (at 5.6 min) with fragments *m*/*z* 153 and *m*/*z* 109, characteristics of protocatechuic acid, thus resulting by a likely hexose group moiety loss; so, it was supposed to be an hexoside derivative of protocatechuic acid [39].

#### 3.2.2. Sauvignon Blanc Wines 

Figure 4 shows the MFA for the Sauvignon Blanc wines that were also computed using the dataset of basic oenological parameters, sensory analysis, volatile compounds, and phenolic compounds. MFA was used to fuse the different datasets in a single statistical computation to give an overview of the correlations between the variables and trends among the observations, as already shown for Chardonnay. As seen in Figure 4A and Figure 3B, the partial axes built on the first two components showed a total variance of 33%. It is observed that the wines from New Zealand are separated due to the contribution of volatile compound-F1 and visual-F2. Similarly, wines from Argentina are separated due to the contribution of the taste-F2 and flavour-F2 components, and olfactory-F1. Wines from Chile are separated due to the contribution of phenolic compounds-F2 component and basic oenological parameters-F2. Furthermore, the wines from Italy are anti-correlated with respect to the taste-F2 and flavour-F2 components, and olfactory-F1. In the case of Sauvignon Blanc, the pH and total acidity (TA) were perfectly anti-correlated. From the loadings plot in Figure 4D, it is possible to see that the olfactory attributes of “passion fruit” and “stone fruit” were highly correlated with OQJ. Wines that were positively characterised by these attributes correlating with OQJ were from Chile and Australia. On the contrary, the olfactory attribute “green bell pepper”, “citrus” flavour, and “warmness” were anti-correlated with OQJ; the wines characterised by these attributes were from South Africa and Argentina. Furthermore, the volatile compound α-terpineol (XII) is known for enhancing the perception of “floral” and “sweet” aromas, and it was shown to contribute to the complexity, and to be one of the key odorants in Sauvignon Blanc wines [10]; 3-mercapto-hexanol (III) was shown to be the main volatile thiol present in Sauvignon Blanc; it is known for being responsible for the “tropical fruit” aroma [43]; ethyl cinnamate (IX) compounds, which were reported to be present by [44], are known for the perception of “sweet”, “balsamic”, “fruity”, “spicy” aromas and were also herein correlated with OQJ. The phenolic compounds x.86 (*m*/*z* 229.1 at 6.6 min) and x.59 (*m*/*z* 311.1 at 5.2 min) showed a correlation with the “sourness” attribute that characterises the wines from New Zealand, while compound x.92 (*m*/*z* 157.1 at 7.0 min) showed a correlation with the olfactory attribute “dried fruit” characterising wines from Northern Italy. Compounds x.54 (*m*/*z* 487.1 at 5.1 min), x.41 (*m*/*z* 230.0 at 4.0 min), x.29 (*m*/*z* 287.1 at 2.9 min), and x.84 (*m*/*z* 315.1 at 6.5 min) showed correlation with wines from Argentina, while wines from South Africa (Western Cape) were more correlated to compounds x.75 (*m*/*z* 373.1 at 5.9 min), x.127 (*m*/*z* 303.1 at 10.7 min), and x.71 (*m*/*z* 175.1 at 5.7 min). The wines from Southern Italy were well explained by dimension 2, which was characterised by compounds x.126 (*m*/*z* 478.2 at 9.6 min) and x.101 (*m*/*z* 325.1 at 7.5 min).

The mass spectrum of the x.59 (*m*/*z* 311.1) showed fragment ions *m*/*z* 179 and *m*/*z* 149 corresponding to the caffeoyl and tartaric acid moieties, respectively, that are characteristic ions of the structure of this compound. Tentatively, this fragmentation pattern was assigned to a derivative of caftaric acid [45]. Additionally, the molecular ion *m*/*z* 487 of the x.54 species was characterised by fragment ions *m*/*z* 311 and *m*/*z* 149, and was therefore speculated to be a derivative of caftaric acid as well [40].

x.92 showed a product ion *m*/*z* 157 co-eluting with *m*/*z* 173 in full MS mode. Additionally, a fragment ion *m*/*z* 113 was found. This compound was tentatively attributed to a derivative of dehydroascorbic acid [46].

The x.54 specie at 5.1 min had a suspected in-source precursor ion *m*/*z* 487, and the fragmentation pattern was characterised by fragment ions at *m*/*z* 355, *m*/*z* 311, and *m*/*z* 149. The fragment ion *m*/*z* 355 was formed by the loss of the pentose unit (132 Da) from *m*/*z* 478. Alternatively, *m*/*z* 487 was hypothesised to be a glucuronyl unit-containing compound (loss of 176 Da), showing the fragment ion *m*/*z* 311 in MS2 [47]. Additionally, *m*/*z* 487 appeared to be composed of caffeic and tartaric acid moiety, as it showed two characteristic product ions *m*/*z* 311 (caffeoyltartaric acid) and *m*/*z* 149 (tartaric acid), and a loss of a caffeic acid moiety (162 Da) [48].

The full MS and MS^2^ spectra for the species x.41 showed a peculiar pattern (fragment ions *m*/*z* 186 and *m*/*z* 142, and a precursor ion *m*/*z* 230, in which two sequential losses of CO_2_ (44 Da) are observed. Thus, compound x.41 possibly included two carboxylic/carboxylate groups, and an odd number of nitrogen molecules; this observation could back up the N-acetyl-hexose-hexose precursor compound tentative assignment [49]. Whereas the species x.41 was tentatively identified as an amino acid derivative; notably, the fragment ions *m*/*z* 186 and 142 could be assigned to a tryptophan derivative, as reported by [36].

The species x.29 had a parent ion *m*/*z* 287, and it was tentatively identified as a flavanone [50]. Notably, its MS^2^ fragmentation pattern reported a fragment ion *m*/*z* 269, produced through a neutral loss of 18 Da from the parent ion corresponding to the loss of water, thus supporting this hypothesis.

The species *m*/*z* 315.1 showed two peaks, one eluted at 4.2 min (x.84) and the other at 6.5 min. It was speculated that the ion *m*/*z* 315.1 at 4.2 min was a derivative of isorhamnetin after the neutral losses 162 Da, corresponding to a hexoxide moiety [51]. Additionally, for *m*/*z* 315.1 at 6.6 min, fragment ions *m*/*z* 271 and *m*/*z* 188 were detected, again indicating a potential isorhamnetin derivative. The fragment ion *m*/*z* 271 was characteristic of isorhamnetin derivative, whereas the fragment ion *m*/*z* 188 was unknown.

The species x.75 (*m*/*z* 373.1) showed a fragment ion *m*/*z* 175 in MS^2^; moreover, in full MS mode further fragment ions *m*/*z* 157, *m*/*z* 131 and *m*/*z* 115 were observed. It is worth noting that the x.75 shared its fragmentation pattern with ascorbic acid (*m*/*z* 175, *m*/*z* 157, *m*/*z* 115) [52]. However, it was not possible to identify the structure of the parent ion *m*/*z* 373.1. Looking at the retention times, besides x.75 specie, an analogue/isomer *m*/*z* 373.1 was eluted at 6.0 min. The first peak (5.7 min) showed a fragment ion *m*/*z* 327, *m*/*z* 193, *m*/*z* 175, and the second peak at 6.0 min released fragment ion *m*/*z* 175.

The ion *m*/*z* 175 was observed also in full MS scan at 5.7 min, as an in-source fragment of *m*/*z* 373.1. Notably, the MS^2^ of *m*/*z* 175 at 5.7 min (an attempt to conduct an indirect MS^3^ experiment on this product ion, by selecting it as an in-source fragment) showed product ions at *m*/*z* 129 (loss of formic acid from the precursor ion) and *m*/*z* 85 (possible loss of CO_2_ from the *m*/*z* 129 product ion itself). The peak at 6.0 min showed fragment ions *m*/*z* 157 (loss of water), 115 (loss of acetic acid) from the precursor ion, and *m*/*z* 85. In full MS mode, the species x.75 also showed *m*/*z* 571, *m*/*z* 373, and *m*/*z* 359 at 5.7 min, whereas *m*/*z* 571 and *m*/*z* 373.1 were observed at 6.0 min, thus confirming that x.71 and x.75 shared similar features (in full MS and MS^2^ spectra) with compounds such ascorbic acid and quinic acid [53]; however, ascorbic acid and quinic acid are too highly polar species to be suitable candidates for these observed features: for example, the injected ascorbic acid standard eluted before 3 min with this chromatographic method; therefore, a definitive assignment for x.75 and its analogue at 6.0 min could not be provided herein.

The x.127 specie at 10.7 min was characterised by fragment ions *m*/*z* 285, *m*/*z* 177, and *m*/*z* 125; the loss of a water molecule (18 Da) produced the fragment ion *m*/*z* 285, whereas *m*/*z* 177, and *m*/*z* 125 were reported to be the fragment ions generated from the cleavage of the C ring of a dihydroflavonol (i.e., a flavanonol). As reported by Shen et al. and in previous papers by our reserch group [6,29,54], this compound is probably identified with taxifolin.

The ion *m*/*z* 325 was found to elute at 7.5 min (x.101) and 7.6 min, thus indicating the presence of two isomers or two derivatives. These assignments were based on its parent ion (*m*/*z* 325), that released, as fragment ions, *m*/*z* 193, *m*/*z* 163 (loss of OCH_3_/coumaroyl fragment), and *m*/*z* 149 at 7.5 min, whereas fragment ions *m*/*z* 193, 113, 103, 87 were at 7.6 min. Ion *m*/*z* 193 is assignable to ferulic acid; in full MS mode, *m*/*z* 193, *m*/*z* 179 (released of caffeate unit), and *m*/*z* 135 were found [55]. Thus, the two species with *m*/*z* 325 at 7.5 min and 7.6 min were tentatively assigned to *cis*- and *trans*-feruloyltartaric acids, respectively.

### 3.3. Regression Models 

#### 3.3.1. Partial Least Squares Regression (PLS-R) for the OQJ Score of Chardonnay Wines

The PLS1 regression method was used to study the combination of sensory attributes that have the greatest impact on the overall quality judgement (OQJ) of Chardonnay wines. The sensory attributes were treated as an X-matrix, while the Y-vector was OQJ. The PLS for OQJ had an R^2^ of 0.85 and an RSME of 0.174. Figure 5A shows the VIP (variables important in the projection). The full regression model is reported in the Appendix A. The variables with a positive contribution in the regression model (Figure 5B) were the “colour green”, olfactory “citrus fruit”, olfactory “pome tree” fruit, olfactory “stone fruit”, olfactory “flowers”, “warmness”, “tropical fruit” flavour, and “woody” flavour. Whereas the variables with a negative effect on the regression were “colour yellow”, olfactory “vegetables”, “sourness”, and “citrus” flavour. The last Figure 5C shows the experimental observations of the overall quality judgment against the model predictions, which show no observation outside the confidence intervals (built at a 95% confidence level).

#### 3.3.2. Multiple Linear Regression to Identify the Volatile Compounds Responsible for the Flavour Sensory Attributes of Chardonnay Wines

Multiple linear regression (MLR) was used to understand which volatile compounds affected the flavour sensory attributes. In Table 6, the indices for the regression of the flavour sensory attributes were reported. It is possible to see that the “tropical fruit” regression had an R^2^ of 0.86 and “woody” an R^2^ of 0.96, while the adjusted R^2^ were 0.81 and 0.95, respectively. Figure 6B shows the predicted vs. experimental data for the “tropical fruit” flavour; only one validation point was outside the confidence interval, while Figure 6D shows the same graph for the “woody” flavour. In this case, none of the data was outside the confidence interval. Figure 6A,C showed the effect of the standardised coefficient on the regression for “tropical fruit” and “woody” flavour, respectively. In the case of “tropical fruit”, the positive effects were given by XXXII, IX (2-pentanol, 3-methyl) and X (butyl lactate), instead, the negative effect was given by XXXV (2,4-di-tert-butylphenol). For the woody flavour, the coefficients that showed a positive effect were X (butyl lactate) and XXXV (2,4 -di-tert-butylphenol), while the negative effect was given by XXXII and X (butyl lactate).

The full regression model is reported in Appendix A.

#### 3.3.3. PLS-R for Visual and Gustatory Data for Chardonnay Wines

For the visual and gustatory data, the PLS regression was used to understand which phenolic compounds had an impact on the sensory attributes. The attributes were “green colour”, “yellow colour”, “warmness” and “sourness” used as the Y-matrix, and phenolic compounds obtained by LC-MS as the X-matrix. The colour green showed an R^2^ of 0.79 and an RMSE of 0.12, the colour “yellow” an R^2^ of 0.65 and RMSE of 0.31; whereas “warmness” showed an R^2^ of 0.73 and RMSE of 0.17; a 0.71 R^2^ and 0.22 RMSE were registered for “sourness” (all the indexes are shown in Table 7). Figure 7 shows the predicted vs. experimental data for the different visual and gustatory attributes. All the data related to the colour “green” and “warmness” were inside the confidence interval; both the colour “yellow” and “sourness” had one data point that was outside of the confidence interval. Additionally, Appendix A shows the VIPs, which were in order of importance: x.11, x.68, x.115, x.96, x.169. Appendix A shows the standardised coefficient effect in the regression.

The full regression model is reported in Appendix A.

#### 3.3.4. PLS-R for the Overall Quality Score of Sauvignon Blanc Wines

To understand the highest impacting sensory attribute on the overall quality judgment (OQJ) of Sauvignon Blanc wines, a PLS regression method was applied. As for Chardonnay, the X-matrix (explanatory variables) contained the sensory attributes, while the OQJ was treated as the Y-vector (dependent variable). The quality of the PLSR was defined by an R^2^ of 0.83 and an RMSE of 0.19. Figure 8A shows the VIP for the OQJ. The most important variables with a positive effect on the regression were passion fruit, stone fruit, and sweetness, while the variables that had a negative effect were woody, warmness, and yellow colour (Figure 8B). Finally, Figure 8C shows the overall quality judgment observations vs. the predictions plot. No data were found outside the confidence intervals (α = 95%).

The full regression model is reported in Appendix A.

#### 3.3.5. PLS-R for the Olfactory Profile of Sauvignon Blanc Wines

As for Chardonnay wines, the Sauvignon olfactory data were computed in the PLS-R against the volatile compounds to understand which of them had the greatest effect on the olfactory sensory attributes. A PLS2 regression model (Y-matrix with included multiple descriptors) was computed for all the olfactory variables but only those that showed an R^2^ higher than 0.6 were reported.

Figure 9A shows the PLS2 model VIP graph. The most important volatile variables were III (3-mercapto-hexanol), VI (acetic acid, hexyl ester), X (furfuryl mercaptan), and XI (trans-2-nonenal). Figure 9B–I show the regression graph of predicted OQJ values vs. the experimentally measured OQJ values. The olfactory variables “grapefruit” (R^2^ of 0.77 and RMSE of 0.20), “passion fruit” (R^2^ of 0.79 and RMSE of 0.22), “woody” (R^2^ of 0.66 and RMSE of 0.31), “vanilla” (R^2^ of 0.64 and RMSE of 0.28), and “green bell pepper” (R^2^ of 0.66 and RMSE of 0.344) showed 6% of the data outside the confidence intervals. Notably, the olfactory variables “tomato stem” (R^2^ of 0.63 and RMSE of 0.32), “flowers” (R^2^ of 0.63 and RMSE of 0.26), and “honey” (R^2^ of 0.72 and RMSE of 0.17) showed no data outside the confidence interval. In Table 8, R^2^, standard deviation, MSE, and RMSE are reported for each of the olfactory variables used in the regression.

It is important to recall that PLS2 takes the whole of the sensory profile as the Y-matrix, thus mediating the effects of volatile profiles on specific descriptors. Consequently, the contributions of each compound to each descriptor still depends on the global sensory profile as a whole. “Grapefruit citrus-a” appears negatively influenced by 3-mercapto-hexyl acetate, acetic acid hexyl ester, butanoic acid ethyl ester, and furfuryl mercaptan, while a remarkable positive effect was given by benzaldehyde. “Passion fruit-a” appears negatively influenced by *trans*-2-nonenal, 1-octen-3-ol and 2-sec-buthyl-3-methoxypyrazine, while the positive effect was given by acetic acid, hexyl ester, benzaldehyde and α-terpineol. The “Woody-a” descriptor appears hindered by benzaldehyde, whereas the highest positive effect was given by 4-mercapto-4-methyl-2-pentanone. “Vanilla-a” was hindered by 2-sec-butyl-3-methoxypyrazine, trans-2-nonenal, and β-damascenone, whereas the highest positive effect was due to acetic acid, hexyl ester followed by butanoic acid, ethyl ester, and 3-mercapto-hexyl acetate. “Tomato stem-a” and “Green bell pepper-a” showed similar effects, as shown by the one-way ANOVA. The highest negative effect was given by 3-mercapto-hexyl acetate followed by furfuryl mercaptan, acetic acid, hexyl ester, butanoic acid, ethyl ester and 2-sec-buthyl-3-methoxypyrazine, while the positive effect was given by 4-mercapto-4-methyl-2-pentanone, α-terpineol and β-damascenone. “Flower-a” was negatively influenced by 1-octen-3-ol and 4-mercapto-4-methyl-2-pentanone, while many compounds had positive effects, such as 3-mercapto-hexyl acetate followed by acetic acid, hexyl acetate, furfuryl mercaptan, benzaldehyde, butanoic acid, ethyl ester, and ethyl cinnamate. The last aroma descriptor, “Honey-a” was negatively affected by β-damascenone, α-terpineol and trans-2-nonenal, whereas the main positive contributions were given by furfuryl mercaptan, ethyl cinnamate, butanoic acid, ethyl ester, and acetic acid, hexyl ester.

The full regression model is reported in Appendix A.

#### 3.3.6. Multiple Linear Regression for the Flavours of Sauvignon Blanc Wines

As for the flavour sensory attributes of Chardonnay wines, the MLR was used to find which volatile compound most affected and predicted the flavour sensory attributes (citrus fruit and woody) in Sauvignon Blanc. In Table 9, the quality index for MLR, on the descriptor citrus fruit flavour, which has an R^2^ of 0.89, is reported. Figure 10A shows the standardised coefficient effect on the regression for citrus fruit flavour. While Figure 10B shows the predicted vs. experimental data in the regression for the flavour analysed. As it is possible to see in Figure 10B, citrus fruit showed one validation data point outside the confidence interval.

The full regression model is reported in Appendix A.

### 3.4. Proanthocyanidins

The list of observed proanthocyanidins (PAC) is reported in Appendix A for both Chardonnay and Sauvignon Blanc [11,56]. The list of observed PAC is reported with their respective used abbreviation.

For both Chardonnay and Sauvignon Blanc, percentage ratios (%) of the cyclic tetrameric procyanidin (%C-4), cyclic tetrameric prodelphinidin having one (epi)gallocatechin and three (epi)catechins (%C-4-OH), and cyclic pentameric procyanidin (%C-5) were calculated [11]. Briefly, %C-4 expresses the ratio of the relative abundance of the measured cyclic tetrameric procyanidin at *m*/*z* 1153.3) [11] over the sum of all cyclic and non-cyclic tetrameric procyanidins *(m*/*z* 1155.3 and *m*/*z* 1153.3)*;* thus, %C-4 represents the percentage of cyclic procyanidin tetramer over the totality of tetrameric procyanidins. Similarly, %C-4-OH is the ratio of cyclic tetrameric prodelphinidin over the sum of cyclic and non-cyclic tetrameric prodelphinidins, and %C-5 is the ratio of cyclic pentameric procyanidins over the sum of cyclic and non–cyclic pentameric procyanidins. The % ratios can provide a normalised parameter for assessing wine authenticity, according to the previous literature [11,56], but in this case, they were applied to test for the different geographical origins, and if these have a significant effect on the profile of PAC. Firstly, an ANOVA test was performed on the Chardonnay and Sauvignon Blanc PAC datasets, separately. Only the significant variables for the origin, shown in Appendix A (*p*-val. < 0.05), were used in a PCA model built to observe the correlations between the variables and samples.

ANOVA was used to identify which of these variables were the most significant, and then apply them in the PCA. Additionally, the use of the %-ratios as supplementary variables was tested, so that their effect would not affect the other variables, being that they are derived from a % calculation of the same variables. Figure 11A reports the PCA for the Chardonnay proanthocyanidins dataset, where only the ratio %C-5 was significant in the ANOVA to differentiate the origin and reported in PCA, as supplementary variables. Instead, in Figure 11B, the PCA for the Sauvignon Blanc proanthocyanidins dataset was reported. In this case, all the %ratios were used because they were found significant by ANOVA for the origin. The PCA for Chardonnay had an explained variance of 71.4%, while Sauvignon Blanc had an explained variance of 69.7%. The PCA for the Chardonnay wines showed one group where almost all the samples are clustered, and only two samples were outside of this cluster: RSA_WC and ARG_ME. As for the Chardonnay, PCA for the Sauvignon Blanc showed one group of samples clustered together; only three samples were outside the cluster: AUS_SE, ITA_SP, and NZL_MR.

Then, ANOVA was run with the combined dataset of Chardonnay and Sauvignon to find out whether the variables were able to distinguish the grape variety, disregarding the origin. The cyclic compounds and ratios that distinguished the grape variety were prodelphinidin (PD) dimers 2-OH (x.7), c-OH tetramer (x.14), procyanidin (PC) pentamers (x.19), and %C4, as these showed a statistical significance in the ANOVA (Table 10).

These variables (dimers 2-OH–PD (x.7), c-OH tetramers–PD (x.14), pentamers-PC (x.19), and the %ratio C-4) were used to build the PCA reported in Figure 12 with Chardonnay and Sauvignon Blanc data. As it is shown in the PCA plots of Figure 12A,B, there was a clear separation between the different varieties. The PCA model showed 94% of the total variance explained in the first three components.

In terms of cyclic proanthocyanidins, the variability in Chardonnay is much lower than that in Sauvignon Blanc. The separation takes place mostly along PC2. Also, another interesting point to note is that the variable that significantly influenced the separation of the two grape varieties was the percentage of the cyclic compound, independently of the geographical origin.

A multivariate discriminant model was also built, to see if the obtained four significant variables could allow a classification of the varieties and the wines in different groups regardless of origin. According to the discriminant analysis built on the significant variables, a 91% accuracy in cross-validation, and a 100% in validation (part of the dataset was applied in validation and not in model training), for discrimination of Chardonnay and Sauvignon (shown in Appendix A) was found, based on the selected significant variables. Thus, cyclic, and non-cyclic proanthocyanidins could be employed to a satisfying extent as varietal markers, independently of the origin; still, the origin of the wines (country~producers) did in fact influence most of the PAC profile (in Appendix A), so the different factors (variety and origin) did leave different signatures on the PAC fingerprint. Notably, a second model was built upon significant variables only, which showed a relatively good classification of the origin, regardless of the grape variety. Appendix A shows the canonical space for the classification of varieties, instead, Appendix A shows the canonical space for the classification of origin. This latter effect, however, cannot be easily explained and will require more investigation.

## 4. Discussion

The first part of this article is based on the Analysis of the Variance (ANOVA) and explorative methods such as AHC and MFA. For Chardonnay wines, MFA showed a correlation of “green colour” with OQJ, also a significant variable in ANOVA; the same was true for “citrus fruit” (aroma) and “tropical fruit” (flavour). Like the sensory variables, some phenolic and volatile compounds also showed correlations with OQJ and were significant in the ANOVA model. Volatile variables that correlated with the OQJ were XIX, XXIX (benzene-butanal), XXVI (acetic acid, 2-phenylethyl ester), and XXXV (2,4-di-tert-butylphenol); x.29, x.123, and x.143 for the phenolic compounds showed correlation with the OQJ. x.29 showed a parent ion *m*/*z* 181.1 that was assigned to dihydrocaffeic acid (fragments *m*/*z* 121 and *m*/*z* 109 also were found). In detail, the fragment *m*/*z* 121 was attributed to the quinone derivative, and the fragment was shown to be probably a deprotonated catechol (*m*/*z* 109). Additionally, the formation of fragment ions *m*/*z* 79, *m*/*z* 80, and *m*/*z* 81 were tentatively identified as derived radical ions [57].

The mass spectrum of x.123 produced a distinctive parent ion *m*/*z* 179.1, and it was tentatively identified based on its fragmentation pattern (fragments *m*/*z* 135, *m*/*z* 134, *m*/*z* 117, *m*/*z* 107, *m*/*z* 89), characteristic of caffeic acid. 

Furthermore, for Sauvignon Blanc, the sensory data that showed correlation with OQJ were “passion fruit” and “stone fruit” aromas, while “tomato stem”, “green bell pepper” (aromas), and “green colour” showed anti-correlation with OQJ and significance in the ANOVA. The volatile and phenolic compounds that showed significance according to ANOVA and correlated in the MFA with OQJ were III (3-mercapto-hexanol), IX (ethyl-cinnamate), and XII (α-terpineol) for the volatile compound, and x.19, x.39 and x.97 for the phenolic compounds.

The x.19 specie at 1.8 min had *m*/*z* 391 as a precursor ion in negative mode and a fragment ion *m*/*z* 241 in product ion mode. According to its fragments, this compound was tentatively assigned to an analogue or a related species of aucubin [58]. Aucubin is an iridoid whose loss of 150 Da was tentatively attributed to a xylose. Additionally, the full MS spectrum showed the characteristic product ions of gallic acid (*m*/*z* 169 and *m*/*z* 125). Notably, in the cited reference, the iridoid was present either in free form or esterified with a phenolic acid [58], potentially indicating the presence of an actual precursor at even higher values than *m*/*z* 391.

The x.39 specie at 3.8 min was characterized by an observed precursor ion *m*/*z* 153 and a fragment ion *m*/*z* 109 in product ion mode. This mass spectrum would be characteristic of protocatechuic acid, but the presence of in-source fragments of an hexoside might indicate that x.39 is a protocatechuic acid hexoside derivative, instead. Nevertheless, it was not possible to identify the precursors ion in the full MS mode.

The x.97 specie showed a fragment ion *m*/*z* 176.7 in product ion mode, although only a small fragmentation was achieved with the applied collision energy. The mass spectrum in full MS mode was characterised by the fragment ions *m*/*z* 179 and *m*/*z* 135, which should be derived by the loss of 44 Da (CO_2_), and they were characteristic of a caffeoyl moiety [59]. Based on the results reported by Šuković et al. [40], x.97 was tentatively identified as a caffeoyltartaric acid derivative; nevertheless, it was not possible to identify the precursor ion in the full MS mode.

A qualitative comparison between the results from the assessment given by our panel, which reported quality as OQJ scores, and the German panel, which reported quality as OWS (overall wine sensory score), allowed interesting observations. The spider plots in Appendix A show a potential discrepancy between the evaluation of the two panels towards the wines. For Sauvignon Blanc, the German panel gave higher scores to the sensory profiles expressed by Argentinian and South African (coastal) products (Sauvignon or Chardonnay), while the Italian panel gave higher scores to the Australian (south-east) and Chilean (central valley) wines. In other words, the Italian panel gave the highest OQJ scores to Sauvignon Blanc with higher “fruity”, and “tropical” notes, and the lowest score to the wines described as “woody”, “grassy” (“vegetative”) and “citric”; the German panel instead assigned the highest OWS (overall wine sensory scores) to wines with the latter characteristics. However, an examination of the effects of the origin and regional composition of the panel on sensory evaluation should be systematically conducted in subsequent studies, including extension to other geographical areas, to reach more solid and scientifically thorough conclusions.

The trend for Chardonnay, on the other hand, was similar, as all wines with the highest OQJ scores given by the German panel were compared with those of the Italian panel (Appendix A, bottom graph); in any case, this seems to be related to the high vegetative and woody flavour of the Argentinian and South African (coastal) products, as was also the case for Sauvignon Blanc wines. This was confirmed by the fact that the German panel also gave the highest scores to the Italian Chardonnay products, which were characterised by a more citric flavour.

These preliminary observations might provide crucial information that could help wineries and wine traders to optimise their commercial strategies on the different acceptance behaviour of their customers worldwide. Of course, as stated, such an application is by no means presented in this work. It requires well-devised ring tests and a much more thorough investigation involving systematic comparisons between the panels’ outputs and validations and ensuring harmonization and standardization of the analysis methods employed. Additionally, a wider evaluation including more samples, more geographical origins for the wines, and other nationalities represented in the panels, will be important to define whether an actual dependence of preference on nationality or geographical origin can be systematically evaluated. However, by applying the proposed chemosensory approach, not only could be possible to extrapolate differences due to the panel composition or origin but also might be possible to express these differences/trends in terms of analytical data, thereby providing an additional tool to the investigators.

The second part of this work presented the regression models. PLS, PCR and MLR methods were tested, and only the best regression models were eventually reported in the article. As mentioned, (see Section 2), the datasets were used applying a rational separation: sensory analysis attributes vs. OQJ, aroma compounds vs. olfactory and retro-olfactory sensory attributes, and non-volatile phenolic compounds vs. visual and gustatory attributes. These logical separations were described in previous work [6], where different regressions were tested to understand which variables majorly affected the different sensory variables.

The PLS1 regression on the OQJ vs. the sensory variables shows interesting results. In Chardonnay samples, the highest VIPs (over 1) with a positive effect on the regression were “colour green”, “citrus fruit-a (“a” indicates olfactory), “pome tree fruit-a”, “stone fruit-a”, “warmness”, “tropical fruit-f”, and “woody-f” flavour. In the case of Sauvignon Blanc samples, the highest VIPs (over 1) with positive effect were “passion fruit” and “stone fruit aroma”, and “sweetness”, instead “yellow colour”, “woody-a” aroma and “warmness” had a VIP over 1 but with a negative effect coefficient in the regression.

Regarding the proanthocyanidins, the results showed clusters in the two varieties with some differences among the origins. For the Chardonnay, the non-clustered samples were RSA_WC (South Africa, Western Cape) and ARG_ME (Argentina, Mendoza), and for Sauvignon Blanc AUS_SE, ITA_SP and NZL_MR. The PCA obtained with the variables that more characterised the origins showed a clear separation between Chardonnay and Sauvignon Blanc over the geographical origins. Those results confirm that proanthocyanidins, and their derived chemical indexes (ratios between non-cyclic condensed tannins and their macrocyclic analogues) [11,56,60], could be used as chemical markers to differentiate the varieties despite the different origins [11].

In conclusion, different methods were investigated to study the quality of these wines. The chemical and sensory analysis along with multivariate statistics were performed first, to understand which variables most impacted the different sensory descriptors. Thereafter, the sensory descriptors, with significant effects on the overall quality judgment (OQJ), were evaluated as the objective answer concerning the quality provided by the trained panel. This multivariate combination of different methods, sensory analysis, and chemical profiling can create a cutting-edge tool to investigate the quality dependence of geographical origins and define an exhaustive overview of the quality of the wines.

Then, defining the overall quality (OQJ) as an objective parameter and statistically defining the attributes most influencing it using regression models, when run in combination with consumers’ surveys, might also give a different perspective on the outcomes of these latter tests. The presence of preferences and prejudices in wine evaluations is a well-known argument, also among experts [61]. Applying regression methods against chemical profiles might provide a sounder background against which evaluations are presented and provide insight into the chemical factors related to viti-enological variables that underpin the given preference scores.

Therefore, this tool may be used by both sellers and traders in the market as an additional objective way to define the quality of their product. To summarise, this approach envisages several future potential applications: (1) to allow validated comparisons between outcomes between different assessors/panels when employing harmonised and standardised methods; (2) to reveal factors inherent to the panel (e.g., the geographical composition) that could skew the assessment results due to higher sensitiveness for one or other compositional traits, as observed here in the comparison between our panels’ outcomes and those of the German panel; (3) thus, to account for such hidden factors and quantitatively describe them as variance trends; (4) in further combination with consumer tests, to also relate results on consumer preferences with descriptive attributes; (5) to build up validated statistical models between the chemical profiles and trends in chemical compositions across viti-enological variables, with the trends observed in the sensory responses of panels or consumers. We must point out that all these applications are envisaged here as potential follow-ups of the present work, and they are not the primary focus of this work; accordingly, they should be addressed systematically in further studies.

## Figures and Tables

**Figure 1 foods-11-03458-f001:**
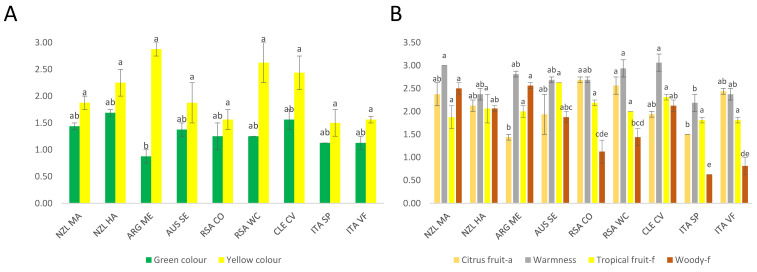
Significant sensory variables according to one-way ANOVA, with the Tukey HSD results. The coloured bars indicate the average between the two duplicates for the different sensory attributes also reporting the standard deviations. (**A**) Significantly different colour attributes, (**B**) Significantly different aroma, taste, and flavour attributes. Letters a–e indicate the groupings found with Tukey’s *post-hoc* analysis.

**Figure 2 foods-11-03458-f002:**
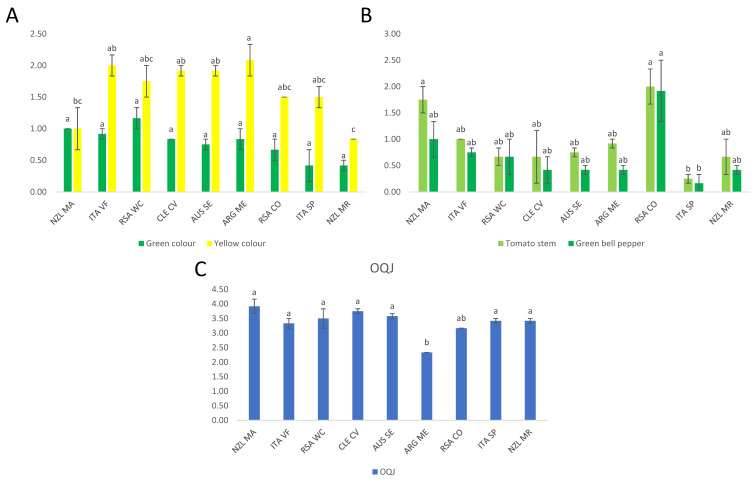
Significant sensory variables according to one-way ANOVA for Sauvignon Blanc data. The bars indicate the average between the two duplicates for the different sensory attributes (also for the standard deviation); different letters show significant differences among groups obtained by the ANOVA. (**A**) Significantly different colour attributes, (**B**) Significantly different aroma attributes. (**C**) Significantly different Overall Quality Judgement (OQJ). Letters a–c indicate groupings found by Tukey’s *post-hoc* analysis.

**Figure 3 foods-11-03458-f003:**
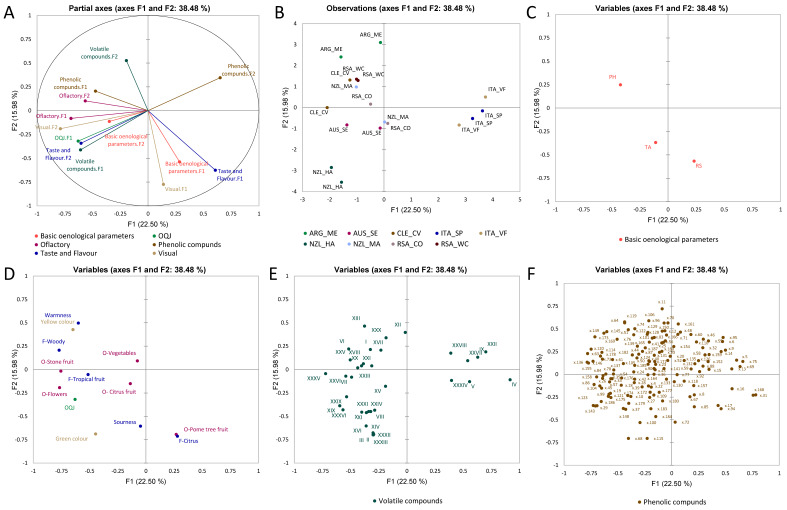
MFA for the Chardonnay dataset. (**A**) shows the partial axes for the first 2 components and the different datasets used in the computation, (**B**) shows the observation plot, and (**C**–**F**) shows basic oenological variables, sensory analysis variables, aroma compounds, and phenolic compounds, respectively. Identified volatile compounds: I = ethyl acetate, IV = 4-amino-1-butanol, VIII = ethylamine, IX = 2-pentanol-3-methyl-, X = butyl lactate, XI = 1-propanol, 3-amino-, XII = 2,3-butanediol, XIII = propanoic acid, 2-hydroxy, ethyl ester or methyl ester, XIV = octanoic, ethyl ester, XV = acetic acid, XVII = 2,4-hexadienoic acid, ethyl ester, XXII = decanoic acid, ethyl ester, XXIII = butanedioic acid, diethyl ester, XXV = naphthalene, 1,2-dihydro-1,1,6-trimethyl-, XXVI = acetic acid, 2-phenylethyl ester, XXVII = hexanoic acid, 2-phenylethyl ester, XXIX = benzene butanal, XXX = phenylethyl alcohol, XXXI = sorbic acid, XXXIII = naphthalene, decahydro-4a-methyl-1-methylene-7-(1-methylethylidene), XXXIV = n-decanoic acid, XXXV = 2,4,-di-tert-butylphenol.

**Figure 4 foods-11-03458-f004:**
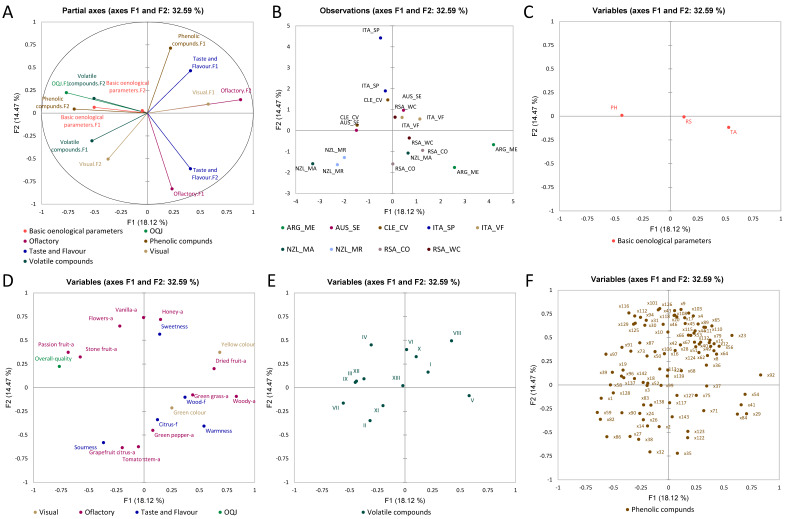
MFA for Sauvignon Blanc dataset. (**A**) shows the partial axes for the first 2 components and different datasets used in the computation. (**B**) shows the observation plot and (**C**–**F**) shows basic oenological variables, sensory analysis variables, aroma compounds, and phenolic compounds, respectively. Volatile compounds: I = 1-octen-3-ol, II = 2-sec-butyl-3-methoxypyrazine, III = 3-mercapto-hexanol, IV = 3-mercapto-hexyl acetate, V = 4-mercapto-4-methyl-2-pentanone, VI = acetic acid, hexyl ester, VII = benzaldehyde, VIII = butanoic acid, ethyl ester, IX = ethyl cinnamate, X = furfuryl mercaptan, XI = trans-2-nonenal, XII = a-terpineol, XIII = b-damascenone.

**Figure 5 foods-11-03458-f005:**
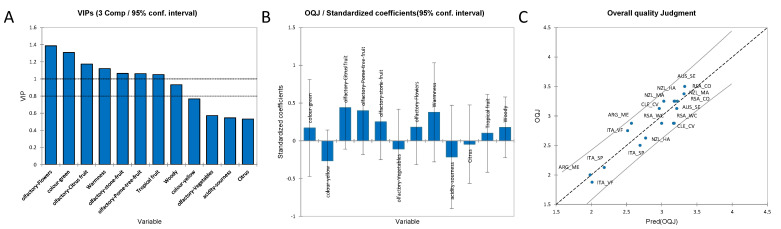
PLS data for the Overall Quality Judgment for the Chardonnay wines. (**A**) shows the VIP for the sensory variables, (**B**) shows the effect of the variables on the PLS equation and (**C**) shows the regression for the OQJ vs. predicted OQJ.

**Figure 6 foods-11-03458-f006:**
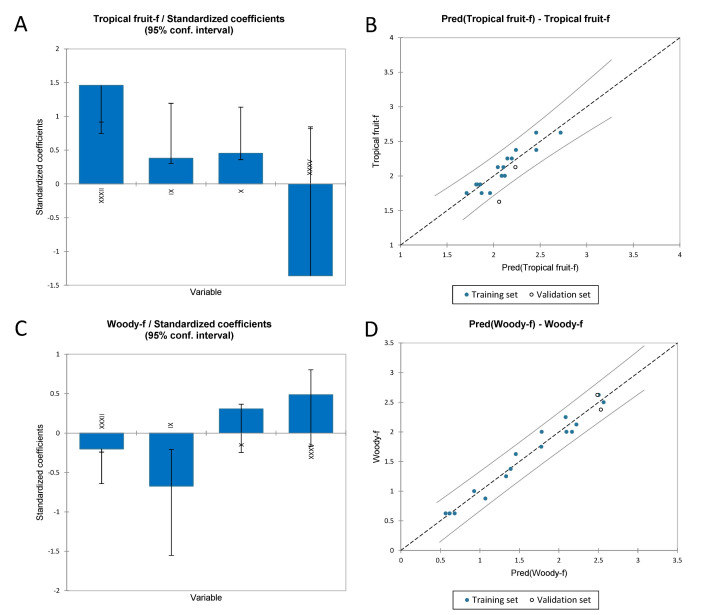
(**A**,**C**) show the standardised coefficients in the multiple linear regression for “tropical fruit” and “woody” flavours on Chardonnay wines. (**B**,**D**) show the predicted vs. experimental data graph for the multiple linear regression on the tropical fruit and woody flavours of Chardonnay wines.

**Figure 7 foods-11-03458-f007:**
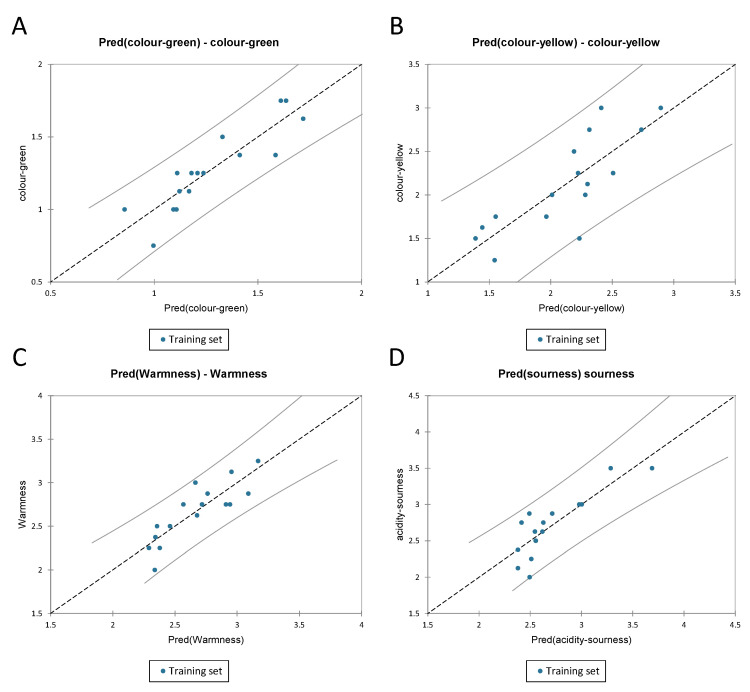
Model-predicted vs. observed visual and gustatory observations for Chardonnay wines. (**A**) predicted vs. experimental data graph for “colour green”, (**B**) predicted vs. experimental data graph for “colour yellow”, (**C**) predicted vs. experimental data graph for “warmness”, and (**D**) predicted vs. experimental data graph for “acidity-sourness”.

**Figure 8 foods-11-03458-f008:**
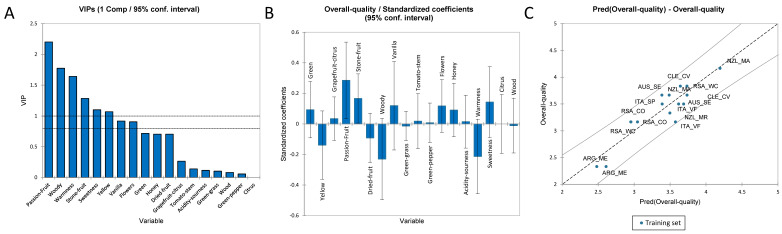
PLS data for the Overall Quality Judgment of Sauvignon Blanc wines. (**A**) shows the VIP for the sensory variables, (**B**) shows the effect of the variables on the PLS equation and (**C**) shows the regression for the OQJ vs. predicted OQJ.

**Figure 9 foods-11-03458-f009:**
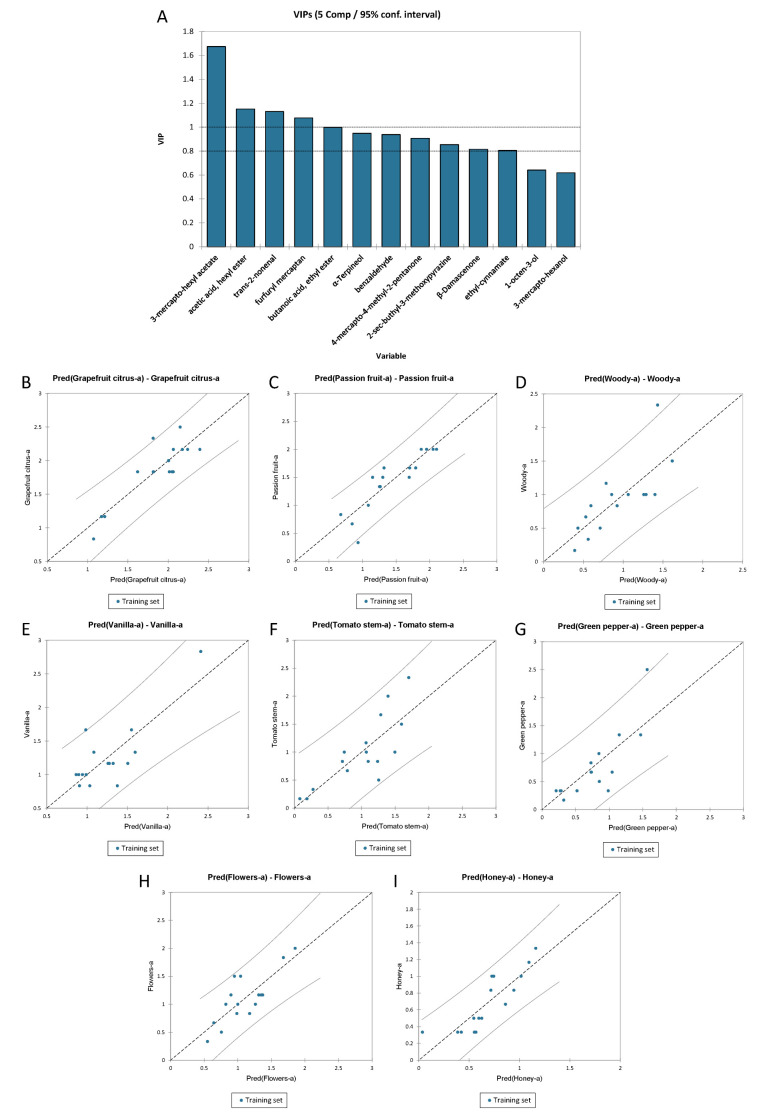
(**A**) shows the variable importance in projection (VIP) in the PLS regression olfactory attributes vs. volatile compounds. Figure from (**B**–**I**) show the regression graphs predicted vs. experiment data for olfactory attributes in the Sauvignon Blanc wines.

**Figure 10 foods-11-03458-f010:**
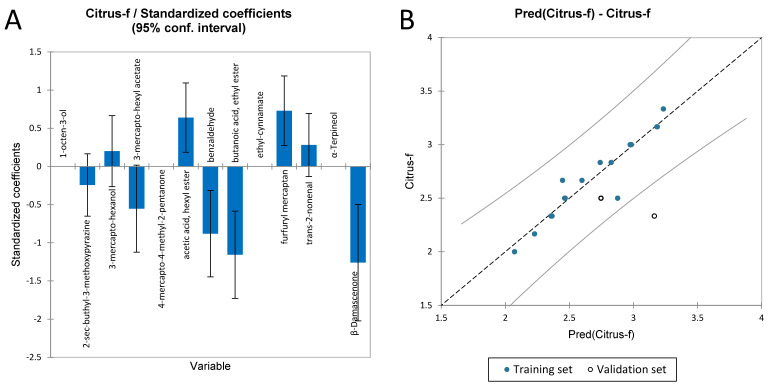
(**A**) Standardised coefficient effect in the multiple linear regression for citrus flavour on Sauvignon Blanc wines. (**B**) Predicted vs. experimental data graph for the multiple linear regression on Sauvignon Blanc wine’s citrus fruit flavour.

**Figure 11 foods-11-03458-f011:**
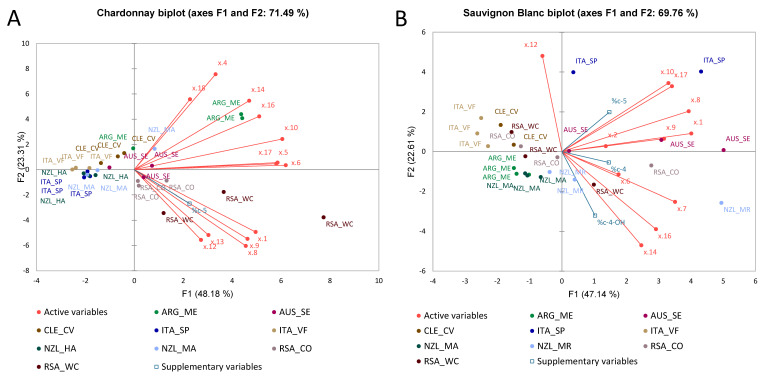
Biplot on PAC data for Chardonnay (**A**) and Sauvignon Blanc (**B**) wines. Supplementary variables (projected in blue) were obtained from the most significant ones in the ANOVA.

**Figure 12 foods-11-03458-f012:**
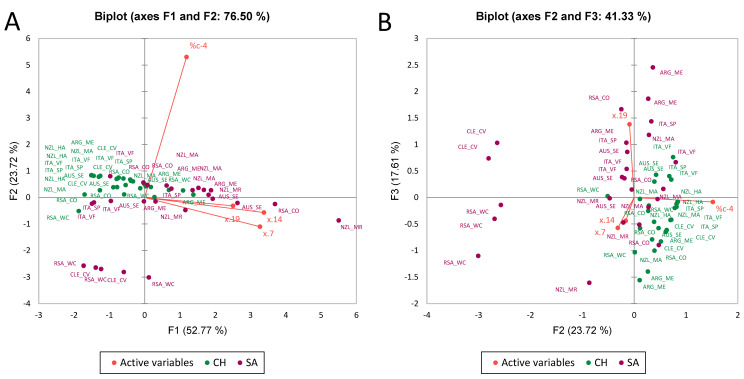
Biplot of PCA for the significant PAC for Chardonnay and Sauvignon Blanc wines. (**A**) Biplot PC1 vs. PC2, (**B**) Biplot PC2 vs. PC3.

**Table 1 foods-11-03458-t001:** Code, origin, and basic oenological parameters of Chardonnay samples (RS, residual sugars; TA, titratable acidity). The alcohol degree (%vol), RS (g/L), and TA (g/L tartaric acid) were analysed by official methods according to OIV [12].

Code	Indication of Origin	Country of Origin	Alcohol (% Vol)	RS (g/L)	TA (g/L Tartaric Acid)	pH	Time and Temperature of Fermentation
CH_NZL_MA	Marlborough (South Island)	New Zealand	14.0	3.2	5	3.5	21 days, 16 °C
CH_NZL_HA	Hawkes Bay (Nord Island)	New Zealand	12.5	3.6	6.7	3.4	14 days, 12–14 °C
CH_ARG_ME	Mendoza	Argentina	14.4	2.3	5.9	3.4	12 days, 13 °C
CH_AUS_SE	South-Eastern Australia	Australia	13.3	4.5	6.1	3.3	25–32 days, 12–18 °C
CH_RSA_CO	Coastal (dry land)	South Africa	13.5	2.1	5.1	3.6	13 °C
CH_RSA_WC	Western Cape (irrigated area)	South Africa	13.2	1.9	5.7	3.8	
CH_CLE_CV	Central Valley	Chile	12.9	1.6	5.6	3.3	10–12 °C
CH_ITA_SP	Sicilia/Puglia	Italy	11.9	3.7	5.2	3.2	8 days, 14–16 °C
CH_ITA_VF	Veneto/Friuli	Italy	12.3	2.8	6.4	3.3	18–20 °C

**Table 2 foods-11-03458-t002:** Code, origin, and basic oenological parameters of Sauvignon Blanc samples (RS, residual sugars; TA, titratable acidity). The alcohol degree (%vol), RS (g/L), and TA (g/L tartaric acid) were analysed by official methods according to OIV [12].

Code	Indication of Origin	Country of Origin	Alcohol (% Vol)	RS (g/L)	TA (g/L Tartaric Acid)	pH	Time and Temperature of Fermentation
SA_NZL_MA	Marlborough (South Island)	New Zealand	13.0	3.8	7.4	3.1	14–21 days, 12 °C
SA_NZL_MR	Martinborough (Nord Island)	New Zealand	12.5	2.5	7.7	2.9	14 days, 12–14 °C
SA_ARG_ME	Mendoza	Argentina	13.0	2.3	6.5	3.3	12 days, 15 °C
SA_AUS_SE	South-Eastern Australia	Australia	11.7	4.41	6.4	3.2	17–21 days, 12–18 °C
SA_RSA_CO	Coastal (dry land)	South Africa	13.5	2.1	5.6	3.4	13 °C
SA_RSA_WC	Western Cape (irrigated area)	South Africa	13.3	1.95	5.43	3.5	
SA_CLE_CV	Central Valley	Chile	13.0	1.7	6.9	3.1	8–10 °C
SA_ITA_SP	Sicilia/Puglia	Italy	12.1	1.1	5.4	3.3	8 days, 14–16 °C
SA_ITA_VF	Veneto/Friuli	Italy	12.9	4.2	6.3	3.3	18–20 °C

**Table 3 foods-11-03458-t003:** Climate details for the different origins of the samples.

Origins	Zone	Climate
New Zealand	Marlborough (South Island)	Warm and temperate. Heavy rainfall even in driest months. Oceanic climate [13].
	Hawkes Bay (North Island)	Mild, generally warm, and temperate with a significant amount of rainfall during the year [14].
	Martinborough (North Island)	Mild, generally warm, and temperate. Significant amount of rainfall during the year [15].
Argentina	Mendoza	Hot and clear summer, cold and cloudy winter, and dry all year round [16].
Australia	South-Eastern Australia	Cool, humid eastern uplands, temperate southeast mallee, inland subtropical northern areas, and hot, dry, arid, and semi-arid country in the far west [17].
South Africa	Coastal (dry land)	Dry, summer-rainfall, relatively warm in winter [18].
	Western Cape (irrigated area)	Mediterranean with warm and dry summers. Mild and moist winters. Low summer rainfall [18].
Chile	Central Valley	Mediterranean climate with cool, dry summers. Mild rainy winters [19].
Italy	Sicilia/Puglia	Mediterranean with hot summers and short mild winters [20]. Mediterranean climate with hot sunny summers and mild winters, dry [21].
	Veneto/Friuli	Moderately continental hill and plain areas; alpine region characterised by cool summers and cold winters with frequent snowfalls [22]. The climate in Friuli Venezia Giulia ranges from the sub-Mediterranean climate of the coastal areas to the more humid temperate climate of the plains, to the Alpine climate in the mountains [23].

**Table 4 foods-11-03458-t004:** List of attributes relevant to the sensory profile of Chardonnay and Sauvignon Blanc wines.

Wine	Attributes	Descriptors	Description
CHARDONNAY	VISUAL		
	GREENISH	From pale grass to an intense green	
	YELLOWISH	From a pale straw to a rich yellow	
	OLFACTORY		
	CITRUS FRUIT-a	Grapefruit, lemon	Aromas from citrus fruit (zest of lemon), particularly grapefruit
	POME TREE FRUIT-a	Green apple, pear	Aromas from pome tree fruit (green apple, pear)
	STONE FRUIT-a	Apricot, peach	Aromas from yellow tree fruit (apricot, peach)
	VEGETATIVE-a	Green bell pepper	Aromas from green bell pepper when cut
	FLORAL-a	Green tea, rose	The fresh scent of wildflowers, jasmine, green tea (infused), and rose water
	CITRUS FRUIT-a	Grapefruit, lemon	Aromas from citrus fruit (zest of lemon), particularly grapefruit
	POME TREE FRUIT-a	Green apple, Pear	Aromas from pome tree fruit (green apple, pear)
	STONE FRUIT-a	Apricot, peach	Aromas from yellow tree fruit (apricot, peach)
	GUSTATORY		
	WARMNESS	Alcohol	Warm/burning sensation in the mouth
	ACIDITY-SOURNESS	Acid (citric, tartaric, malic, lactic)	Acid taste resembling that of vinegar, lemon juice etc.
	BITTERNESS	Caffeine	The bitter taste of caffeine (coffee)
	FLAVOUR		
	CITRUS FRUIT-f	Grapefruit, lemon	The flavour of citrus fruit: grapefruit, orange, lemon zests
	TROPICAL FRUIT-f	Ananas	The flavour of tropical fruit: ananas
	WOODY-f	Wood, oak	The flavour of fresh and toasted wood (oak, resinous aromas)
	CITRUS FRUIT-f	Grapefruit, lemon	The flavour of citrus fruit: grapefruit, orange, lemon zests
SAUVIGNON BLANC	VISUAL		
	GREENISH	From pale grass to an intense green	
	YELLOWISH	From a pale straw to a rich yellow	
	OLFACTORY		
	GRAPEFRUIT CITRUS-a	Grapefruit, citrus zest	Aromas from citrus fruit (zest of lemon), particularly grapefruit
	TROPICAL FRUIT-a	Passion fruit, tropical fruit	Tropical fruit aromas, passion fruit (mango, maracuja)
	STONE FRUIT-a	Apricot, peach	Aromas from three yellow fruit (apricot, peach)
	DRIED FRUIT-a	Nutty, almond	Aromas from dried fruit (nutty, almond, hazelnut, walnut)
	WOODY-a	Wood, oak	The aroma of fresh and toasted wood (oak, resinous aromas)
	VANILLA-a	Vanilla, vanilla sugar	The aroma of vanilla (vanilla sugar, aromas like cake)
	GREEN GRASS	Fresh cut grass	Cut grass aromas or hay
	TOMATO STEM-a	Tomato stem	The aroma of tomato stem
	GREEN BELL PEPPER-a	Green bell pepper	The aroma of green bell pepper when cut
	FLOWERS-a	Wildflowers, jasmine	The fresh scent of wildflowers and jasmine
	HONEY-a	Honey	Aromas of wildflower-honey
	GUSTATORY		
	WARMNESS	Alcohol	Warm/burning sensation in the mouth
	SOURNESS	Acid (citric, tartaric, malic)	The acid taste resembles that of vinegar, lemon juice, etc.
	SWEETNESS	Sucrose. glucose, fructose	Sensation typical of sweet food/beverages
	FLAVOUR		
	CITRUS-f	Orange, grapefruit, lemon and their zest	The flavour of citrus fruit: grapefruit, orange, lemon, and zest.
	WOODY-f	Wood, oak	The flavour of fresh and toasted wood (oak, resinous aromas)
OVERALL QUALITY JUDGEMENT (OQJ)		An objective answer by the panel on the quality of the product	

**Table 5 foods-11-03458-t005:** List of factors considered in the experimental design applied in the HS-SPME method optimisation.

	Extraction Temperature (°C)	Incubation Time (min)	Extraction Time (min)	Fiber Composition
Type	Quantitative	Quantitative	Quantitative	Qualitative
Min level	30	5	10	PDMS Polyacrylate CAR-PDMS DVB-CAR-PDMS
Max level	60	40	60	

**Table 6 foods-11-03458-t006:** Quality indexes of the multiple linear regression models on flavour Chardonnay data.

Statistic	Tropical Fruit	Woody
R^2^	0.866	0.968
Adjusted R^2^	0.817	0.956
MSE	0.016	0.021
RMSE	0.125	0.143
Press	0.593	0.662
Q^2^	0.539	0.905

**Table 7 foods-11-03458-t007:** Quality indexes of PLS regression model on Chardonnay visual and gustatory data.

	Colour Green	Colour Yellow	Warmness	Sourness
R^2^	0.795	0.657	0.730	0.719
Std. deviation	0.149	0.378	0.205	0.265
MSE	0.015	0.098	0.029	0.048
RMSE	0.124	0.314	0.170	0.220

**Table 8 foods-11-03458-t008:** Quality index for PLS regression on aroma descriptors for Sauvignon Blanc.

	Grapefruit Citrus	Passion Fruit	Woody	Vanilla	Tomato Stem	Green Pepper	Flowers	Honey
R^2^	0.771	0.790	0.663	0.640	0.638	0.664	0.639	0.727
Std. deviation	0.264	0.283	0.392	0.363	0.457	0.435	0.329	0.216
MSE	0.044	0.050	0.096	0.083	0.131	0.118	0.067	0.029
RMSE	0.209	0.224	0.310	0.287	0.362	0.344	0.260	0.171

**Table 9 foods-11-03458-t009:** Quality index for MLR on Sauvignon Blanc data.

	Citrus Fruit F
R^2^	0.897
Adjusted R^2^	0.743
MSE	0.038
RMSE	0.195
Press	1.697
Q^2^	0.234

**Table 10 foods-11-03458-t010:** ANOVA results for significant proanthocyanidins (PAC) differentiating Chardonnay and Sauvignon Blanc with the PAC. a and b letters represent groupings computed by Tukey’s *post-hoc* test.

	Dimers 2-OH (PD) x.7	Tetramers c-OH x.14	Pentamers x.20	%c-4
SA	18,298.216 b	4153.848 b	84.447 b	73.348 a
CH	11,328.794 a	3158.361 a	12.081 a	94.812 b
Pr > F(Model)	0.025	0.013	<0.0001	0.009

## Data Availability

Acquired data are available upon request.

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
