# Peer review of "Fusion of 2DGC-MS, HPLC-MS and Sensory Data to Assist Decision-Making in the Marketing of International Monovarietal Chardonnay and Sauvignon Blanc Wines"

_foods, 2022, doi:10.3390/foods11213458_

Round 1
Reviewer 1 Report
This paper proposes a complete oenological analysis (sensory and chemical) of two types of single-varietal white wine (Chardonay and Sauvignon Blanc) from different regions of the world. It studies the quality factors and shows the differences between the two taster panels (one Italian and one German). The main conclusion of the paper is that the differences in appreciation between the nationalities correspond to taste (and therefore chemical) differences. Thus, sellers and producers could adapt their wines according to the markets. This paper is very interesting but suffers from several limitations.
1. The economic literature is absent from the paper. Yet the question of the link between taste, preferences and demand (and thus wine sales and prices) has been widely addressed by several papers (see in particular Oczkowski, 2017). The fact that some wines are better suited to certain markets than others is not new, neither scientifically nor empirically, as importers have always known how to buy the wines that best suit their customers. Moreover, a recent paper has highlighted the biological and cultural origins of preference differences . It is imperative to better contextualise the introduction by referring to this literature.
2. The sensory analysis is very insufficiently described. After re-reading several times, I still cannot find the number of participants in the tasting panels. This number determines the whole empirical analysis. I cannot find this number in the empirical part either. How many observations are there? You can't make statistics without knowing the number of observations, the quality of the work depends on it.
Moreover, nothing is said about representativeness. A panel is very unbalanced between men and women, but we have very little information. Here again, this is crucial to the quality of the results. The results can be biased by poor representativeness (high presence of young people, men, as is the case in the experiment). In order to draw reliable conclusions, it is necessary to have correctly constituted or large panels (to allow the law of large numbers to operate). This is therefore a very strong limitation of the paper.
3. Concerning the geographical origin of the panelists, there is one group of Italians and one of Germans. We are in Bolzano, they are mainly university students if I understand correctly. Bolzano, as far as I know, is in an area of dual German and Italian culture. The differences between the two panels do not seem to me to be sufficient to conduct a study on the differences in preferences between nationalities. We would need more different samples from the major consumer markets for these white wines (USA, UK). This is not critical, but this limitation greatly reduces the scope of the results. It should be emphasised in the text.
4. Regression models should be written so that it is clear which variables are included in the equation. The number of observations in the results tables should also be specified. I would also like to see if individual characteristics have an impact on the preferences / ratings of the panelists. This question needs to be addressed.
References :
Oczkowski, E. (2017). The preferences and prejudices of Australian wine critics. Journal of Wine Research, 28(1), 56-67.
Author Response
We have now replied to the Reviewer’s queries. Please, find below our answers (A) to each specific point raised (Q).
Q.: This paper proposes a complete oenological analysis (sensory and chemical) of two types of single-varietal white wine (Chardonay and Sauvignon Blanc) from different regions of the world. It studies the quality factors and shows the differences between the two taster panels (one Italian and one German). The main conclusion of the paper is that the differences in appreciation between the nationalities correspond to taste (and therefore chemical) differences. Thus, sellers and producers could adapt their wines according to the markets. This paper is very interesting but suffers from several limitations.
A.: We thank the Reviewers for the positive feedback on our work. Regarding the comparison between the two panels, the main aim of the study was to evaluate the wines according to the different origins applying MRATA and analytical methods, not at all a comparison between the two panels. The wines were described in terms of attributes evaluated with a MRATA method, and chemical profiles analyzed by GCxGC and UHPLC.
The main conclusion of the paper was not at all that the differences in preference between the two nationalities corresponded to difference in chemical profiles. We instead first evaluated qualitatively by MFA the overall wines profiles and then more directly correlated 1) the overall quality judgment with the sensory descriptors, and 2) the sensory descriptors (obtained with panelists trained for those specific descriptors) with chemical profiles, applying regression models.
The overall quality we present does not in any way equate with consumers or wine-experts’ preferences. Our study surely had none of these intentions, and a completely different method was employed. The comparison is only qualitative put forward, and briefly shown to indicate a potential effect which would anyway require much more standardized and controlled conditions. In no way, this study claims to have proved it.
Actually, the linking of the results of the German panel with consumer preference is due to the fact that that particular analysis had indeed this aim, but in no way this is purported in the text as a result. It was to us an interesting observation, which we merely noted.
Therefore, we have tried in this new version to make much clearer that we have no intention of presenting the comparison between the two panel as a result of this work.
Please, see the revise version, in which we sidelined all apparent conclusions about this point.
Q.: 1. The economic literature is absent from the paper. Yet the question of the link between taste, preferences and demand (and thus wine sales and prices) has been widely addressed by several papers (see in particular Oczkowski, 2017). The fact that some wines are better suited to certain markets than others is not new, neither scientifically nor empirically, as importers have always known how to buy the wines that best suit their customers. Moreover, a recent paper has highlighted the biological and cultural origins of preference differences . It is imperative to better contextualise the introduction by referring to this literature.
A.: We thank the Reviewer for this comment. We have included new sentences in the Introduction and in the Conclusions, highlighting the presence of preferences and prejudices also among experts. Please, see all changes made to the draft according to this point. In agreement with these changes.
However, as already mentioned in the previous reply, the manuscript’s main aim was to evaluate the overall quality of monovarietal wines across different origins. In no way we wanted to achieve a representative view of consumers’ preferences. As well, our aim was not to correlate economic data with preferences.
In our view, the application of such approach into wine marketing could be an appealing application to get more insight into consumers’ or experts’ preferences, albeit this is not addressed in the manuscript, if not as a potential application. Please, see also the related answers and amendments made in the revised version of the manuscript. In our view, founding/expressing the overall quality as combination of sensory attributes has several appealing applications, all outlined in the manuscript.
Q.: 2. The sensory analysis is very insufficiently described. After re-reading several times, I still cannot find the number of participants in the tasting panels. This number determines the whole empirical analysis. I cannot find this number in the empirical part either. How many observations are there? You can't make statistics without knowing the number of observations, the quality of the work depends on it.
A.: We included some information on the panel composition. The number of observations has been more clearly indicated in the same paragraph.
Q.: Moreover, nothing is said about representativeness. A panel is very unbalanced between men and women, but we have very little information. Here again, this is crucial to the quality of the results. The results can be biased by poor representativeness (high presence of young people, men, as is the case in the experiment). In order to draw reliable conclusions, it is necessary to have correctly constituted or large panels (to allow the law of large numbers to operate). This is therefore a very strong limitation of the paper.
A.: We thank the Reviewer for this insight. The sensory analysis on Sauvignon Blanc conducted in Bolzano (Italy) were performed by students and staff of Italian background, albeit from different North Italian regions. Instead, the sensory analyses on Chardonnay were performed with a group of students that came from all over the world. Four came from Italy, one from Greek, one from USA, one from Japan, and one from Cape Verde. Also, these are trained panel, specifically for the varieties investigated.
We must point out that representativeness for the population (Italian? South Tyrolean?) was not at all sought in this study, as our aim was not to assess consumers’ preference, but define overall quality and attributes on a MRATA protocol with trained panelists, then we aimed at correlating these results with chemical profiles. The main idea was to found sensory attributes on chemical profiles on one side. On the other, correlate the overall quality (measured as an objective parameter) with the attributes. This latter points represent the main aim of our work. In our view, achieving a description of the main attributes The employed panels underwent a specific training for this aim. Again, we fear that the comparison between our panel’s results and those of the German panel was given too much attention. For this reason, those conclusions have now been sidelined. We hope that this might clarify the point even further. Besides, the applied protocol was based on previous reported literature (Nishida et al 2021), with conditions very close to our own ones.
Q.: 3. Concerning the geographical origin of the panelists, there is one group of Italians and one of Germans. We are in Bolzano, they are mainly university students if I understand correctly. Bolzano, as far as I know, is in an area of dual German and Italian culture. The differences between the two panels do not seem to me to be sufficient to conduct a study on the differences in preferences between nationalities. We would need more different samples from the major consumer markets for these white wines (USA, UK). This is not critical, but this limitation greatly reduces the scope of the results. It should be emphasised in the text.
A.: Please, see the information on the panel composition in the previous reply. The mixed culture present in Bolzano scarcely influenced the composition of the panel, as the students were recruited among national and international students. Indeed, really a minority of the participants were from Bolzano itself.
As already indicated, we did not evaluate preferences at all. So, we have included a sentence in the Conclusions highlighting this point. The sentence states: “Besides, wider evaluations including other nationalities might be important in order to test whether an actual dependence of preference on nationality or geographical origin can be evaluated systematically. However, applying the proposed chemosensory approach, not only it could be possible to extrapolate differences due to the panel composition or origin, but also it might be possible to express these differences/trends in term of analytical data, so providing a further tool to the investigators.” Moreover, please see all made changes related to the discussed point.
Q.: 4. Regression models should be written so that it is clear which variables are included in the equation. The number of observations in the results tables should also be specified. I would also like to see if individual characteristics have an impact on the preferences / ratings of the panelists. This question needs to be addressed.
A.: We have included the Regression equations as Tables in a new Supplementary File (File 3 Table 3.1).
The regression models were anyway all fully represented as coefficients’ plots in Figure 5 (B: Standardized coefficients for Overall Quality Judgment model for the Chardonnay wines), Figure 6 (MLR on volatile compounds for Chardonnay: A and C show the standardized coefficients in the multiple linear regression for “tropical fruit” and “woody” flavours on Chardonnay wines), Figure S7 (PLS-R model for Chardonnay on non-volatile phenolic compound for the visual and gustatory data: B to M show the standardized coefficients for the variables in the regression), Figure 8 (B: Standardized coefficients for Overall Quality Judgment model for the Sauvignon wines), Figure 10 (A: standardized coefficient effect in the multiple linear regression for citrus flavour on Sauvignon Blanc wines).
The choice of placing Figures in the main text or in the Supporting Information derived mostly from space requirements and further easing the discussion of them.
Q.: References :
Oczkowski, E. (2017). The preferences and prejudices of Australian wine critics. Journal of Wine Research, 28(1), 56-67.
A.: We thank the Reviewer for sharing this important reference. We have obtained the document and proceeded to cite it in order in the Conclusions in order to better contextualize our main results.

Reviewer 2 Report
I have carefully read the manuscript. The authors describe their attempt to discriminate international monovarietal Chardonnay and Sauvignon Blanc wines based on the combination of sensory and chemical analysis and chemometrics to define their quality and to understand which variables had a major impact on the sensory perceptions. The topic is interesting and deals with meaningful research for the food science and technology area. I recognize that the authors have here an extensive and valuable work. Despite being well written, authors should summarise it or split the manuscript into 2 parts, because it is too extensive and sometimes too repetitive and descriptive. The importance of the study it is well justified, the objectives are clearly stated, the experimental includes appropriated and advanced analyses and the obtained data is extensively discussed. However, the current study seems to fail in the Multiple Factor Analysis, since total variance seems to be quite low, which can mean that the items are not sufficient to explain this model. Thus, you should consider have additional items or revisit the whole process. Additionally, the comments stated in the pdf attached should be considered. For these reasons, I recommend major revisions.

Author Response
We have now replied to the Reviewer’s queries. Please, find below our answers (A) to each specific point raised (Q).
Q.: I have carefully read the manuscript. The authors describe their attempt to discriminate international monovarietal Chardonnay and Sauvignon Blanc wines based on the combination of sensory and chemical analysis and chemometrics to define their quality and to understand which variables had a major impact on the sensory perceptions. The topic is interesting and deals with meaningful research for the food science and technology area. I recognize that the authors have here an extensive and valuable work. Despite being well written, authors should summarise it or split the manuscript into 2 parts, because it is too extensive and sometimes too repetitive and descriptive.
A.: We thank the Reviewer for the positive feedback. Regarding the proposal of splitting the study in two, in our view that could only be done by presenting two reports, one on Chardonnay and one on Sauvignon Blanc. This would on one side undermine the results drawn by applying the condensed tannins profiles. On the other, in our view presenting two sets of monovarietal wines does strengthen the evidence that our approach has value, especially for the application of regression analysis. Besides, we have now made extensive changes due to Reviewers’ observations. We think that all the modifications made have improved the draft and also reduced the length of less relevant parts (e.g. description of hierarchical cluster analysis in Material and Methods).
Q.: The importance of the study it is well justified, the objectives are clearly stated, the experimental includes appropriated and advanced analyses and the obtained data is extensively discussed. However, the current study seems to fail in the Multiple Factor Analysis, since total variance seems to be quite low, which can mean that the items are not sufficient to explain this model. Thus, you should consider have additional items or revisit the whole process. Additionally, the comments stated in the pdf attached should be considered. For these reasons, I recommend major revisions.
A.: We thank the Reviewer for this observation. However, the MFA is in this context used as a qualitative assessment of the samples grouping and relationships with the variables based on contributions from many datasets, all characterized by very different number of variables. We lack a strong set of few dominant factors, for which a minimal explained variance > 60% would be surely a requirement. In our case, MFA was merely useful for a more compact summary of the results, i.e. a method to combine the representations of all datasets in one “economical” visualization.
In comparison, presenting all PCA models would have required 4 x 2 plots (or at least 4 biplots, now presented in SI), for each wine. A discussion would have been then necessary to comment the results of each specific analysis, for each wine. So, MFA offers a way to a more compact representation when used just as a qualitative tool. As said, the lack of a dominant major factor is at the basis of the low % of explained variance, but in our view, this does not undermine the usefulness of the method.
In our view, it could be way more relevant to focus on the presence of trends clearly observable in the Score plots. The presence of clusters is clearly observed, not only among replicates, but also between samples from single nations when multiple origins for the same nations are present (e.g. Italian wines in the Chardonnay score plot, Figure 3B). This should be considered an important result.
When presenting MFA from datasets containing many variables (e.g. mass spectrometric, spectrophotometric, etc.) also other reports in the literature faced MFA results with %Variance lower than 60%, and still gaining useful insight (see for example https://doi.org/10.20870/oeno-one.2021.55.4.4805). The variables number in the analysed datasets in relation to the total number of observations must be considered.Q.: peer-review-23105952.v2.pdf
A.: We thank the Reviewer for the proposed changes. We have included most of the proposed amendments in the revised manuscript. Please, find here below our Replies to some of the specific points (all points not discussed here below have all been addressed directly as changes in the main draft):
1) “Typicality or Typicity?”. A.: The text was amended.
2) “Considering that the wine matrix influences the SPME performance, why did not you select a Chardonnay or Sauvignon Blanc wine for the model wine? You should at least mention the basic oenological parameters of the chosen wine so that one can verify if this wine holds similar characteristics to those of the analyzed samples.”
A.: We are well aware that the choice of the matrix was a strong approximation. However, even using a model wine of each monovarietal wines tested would not ensure optimality (please, see https://doi.org/10.3390/pr9040662) due to the extremely high complexity entailed in the matrix-analytes-SPME interactions. So, albeit the suggested approach might have improved a little the accuracy, we still think our method to be absolutely valid for the specific study we presented. In summary, we opted for a practical approach, with the application of an internal standard and added analytes.
In addition, the major advantage for selecting that specific matrix, was that the employed matrix is a commercial wine that presents really consistent characteristics across all batches and over time, which we think is a strong asset onto which rely, especially when the analyses have to be done at very different points in time (e.g., days, weeks, months) using newly opened bottles for preparing the model solutions.
We have included the info about the main characteristics of the model wine used as matrix in the Material and Methods section.
3) “In Table S2 you have assigned trans-caftaric acid and epicatechin with an asterisk, but you have said that pure standards were available. Check if this is correct.”
A.: The Table has been amended.
4) “Poor resolution. Please, upgrade this figure.”
A.: All figures have been uploaded in high-resolution version. The low resolution in the proofs file is due to the format of the file. Among the uploaded attachments there is also a folder with images in high-resolution.
5) “A different subsection should be included here.”
A.: Amended.
6) “???” in hierarchical clustering analysis section.
A.: AHC acronym is now specified at the beginning of the section.
7) “Is this discussion only based on the first two components? Usually, total variance explained by factor analysis should not be less than 60%. The variance explained by the presented model is too low, suggesting that the data is not meaningful. If so, you might need to revisit your whole process.”
A.: Please, refer to our previous answer.
8) “The tentatively identification of polyphenols is missing.”
A.: All tentative identifications of the polyphenols, when possible, have been reported in the text. It would be too cumbersome to report them all in the Figures’ captions, as done for the volatile compounds, considering we have a high number of phenolic variables.

Round 2
Reviewer 2 Report
The manuscript was substantially revised, and the authors made meaningful changes that considerably improved the manuscript. However, authors should carefully read the revised version of the manuscript once again and correct minor issues related with text editing.